# Bridging The Gap Between Training And Testing For Certified Robustness

## Abstract

Certified robustness provides a theoretical lower bound for adversarial robustness and arouses widespread interest and discussions from the research community. With theoretical support to improve the certified robustness on the training set, practitioners endeavor to train a more certified robust model during inference on the test set. However, the experimental neglect on the training set and the theoretical ignorance during inference on the test set induce a gap between training and testing for certified robustness. By establishing an equivalence between the convergence of training loss and the improvement of certified robustness, we recognize there is a trade-off between expressive power and generalization (assuming a well-conditioned optimization) for certified robustness, which is similar to the underfitting and overfitting discussed in machine learning. To investigate this trade-off, we design a new orthogonal convolution-Controllable Orthogonal Convolution Kernel (COCK), which provides a broader range of expressive power than existing orthogonal convolutions. Empirically, there is a power-driven shift from vanilla classification accuracy to certified robustness in the sense of the optimal trade-off between expressive power and generalization. The experimental results suggest that by carefully improving the expressive power from the optimal trade-off for vanilla classification performance, the model will be more certified robust.

## 1 Introduction

The adversarial robustness (Szegedy et al., 2013) of Deep Neural Networks (DNNs) has attracted extensive attention from the research community. Theoretically, certified robustness serves as a lower bound for adversarial robustness against any adversary under certain constraints (Tsuzuku et al., 2018). Currently, though Multi-Layer Perception (MLP) trained on MNIST (LeCun, 1998) has achieved great success in certified robustness, neither MLP nor even Convolution Neural Networks (CNNs) perform satisfactorily on real-world datasets such as CIFAR (Krizhevsky, 2009) and ImageNet (Russakovsky et al., 2015). In particular, the certified robustness degrades dramatically during inference on the test set.

At present, practitioners are devoted to improving the certified robustness of different network architectures during inference on real-world datasets while most of the studies are establishing theoretical guarantees on the training set. The experimental neglect on the training set and the theoretical ignorance during inference on the test set induce a gap between training and testing for certified robustness. Our paper aims to understand this gap and explore what actually affects the certified robustness during inference rather than during training.

Revisiting the fundamental problem in machine learning that there is a trade-off between expressive power and generalization (overfitting and underfitting) when assuming a well-conditioned optimization,[1] we realize there is a similar trade-off for certified robustness as well by establishing an equivalence between the convergence of classification loss and the improvement of certified robustness for Lipschitz-constrained models. Upon understanding the trade-off for certified robustness, we define the risk and accuracy for certified robustness following the corresponding descriptions in machine learning.

---

[1]For simplicity, we omit this assumption in the following when describing this trade-off.

To explore this trade-off further, we focus on designing a tool capable of altering the expressive power. At present, orthogonal constraints on convolutions (Singla & Feizi (2021); Li et al. (2019); Trockman & Kolter (2021)) for the 1-Lipschitz property have achieved relatively competitive performance in both vanilla classification accuracy and certified robustness on real-world datasets. Nevertheless, owing to the strict orthogonal constraint on the parameter space, orthogonal convolutions lack enough expressive power to cover the training set[2] and the flexibility to alter the expressive power. Motivated by the Orthogonal Newton Iteration (ONI) algorithm (Huang et al., 2020) that trades expressive power and orthogonality off, we design a new orthogonalization convolution kernel-Controllable Orthogonal Convolution Kernel (COCK) to provide a broader range of expressive power by altering the restriction to the parameter space.

Theoretically, COCK is proven to be more certified robust than other general orthogonal convolutions owing to the improvement of expressive power. Empirically, the improvement of expressive power enables COCK to cover the training set better and hence improves the certified robustness on the training set. Both theoretical and empirical results align well with our understanding of the equivalence between classification loss convergence and certified robustness improvement.

By altering the expressive power of COCK, we experimentally observe that the optimal trade-off between expressive power and generalization for certified robustness is more powerful than the optimal trade-off for vanilla classification accuracy during inference. Namely, there is a power-driven shift from vanilla classification accuracy to certified robustness. By carefully improving the expressive power from the optimal trade-off for vanilla classification accuracy, we can obtain a more certified robust model.

We summarize our contributions as follows:

- We recognize there is a gap between training and testing for certified robustness. Further, we identify that there is a trade-off between expressive power and generalization for certified robustness by establishing an equivalence between enough power to cover the training set and the improvement of certified robustness. Similar to the fundamental framework in machine learning, we formulate the risk and accuracy in the setting of certified robustness.

- We design a new orthogonal convolution-Controllable Orthogonal Convolution Kernel (COCK), serving as a tool to investigate the trade-off between expressive power and generalization for certified robustness. COCK alleviates the weakness of orthogonal convolutions in expressive power and provides a broader range of expressive power.

- We empirically observe that the optimal trade-off between expressive power and generalization for certified robustness is more powerful than the optimal trade-off for vanilla classification accuracy. By carefully improving the expressive power from the optimal trade-off for vanilla classification accuracy, we can obtain a more certified robust model.

## 2    RELATED WORK

**Certified Robustness for Lipschitz-constrained Models.** Several studies have been conducted to understand and improve the certified robustness of Lipschitz-constrained models. Tsuzuku et al. (2018) established certified robustness for Lipschitz-constrained neural networks against $l_2$ attack and suggested that the Lipschitz norm and the output margin jointly determine the certified robustness on the training set. Upon this framework, on the one hand, multiple studies (Cisse et al. (2017); Gouk et al. (2020); Miyato et al. (2018); Qian & Wegman (2019)) intend to control the Lipschitz norm of neural networks. On the other hand, optimization techniques (Lee et al. (2020); Ono et al. (2018)) are developed to modify the training process by enlarging the output margin. All these works are devoted to theoretically improving the certified robustness on the training set and empirically improving the certified robustness on the test set, leaving a gap between training and testing to be explored.

**Orthogonal Convolution.** Orthogonalization methods regularizing for the 1-Lipschitz property remarkably improve certified robustness. Generally, it is categorized into soft orthogonal constraint

---

[2]In this paper, we use "cover the training set" to describe the training loss a model can achieve, which considers both expressive power and optimization.

and hard orthogonal constraint. The soft orthogonal constraint (Xie et al. (2017); Balestriero & Baraniuk (2018); Balestriero & richard baraniuk (2018); Lu et al. (2018); Miyato et al. (2018); Yoshida & Miyato (2017)) restricts the parameter space by directly adding an orthogonalization penalty to the loss function during training. The hard orthogonal constraint (Huang et al. (2017); Singla & Feizi (2021); Li et al. (2019); Trockman & Kolter (2021)) usually utilizes reparameterization tricks to restrict the orthogonality directly. However, the former cannot guarantee the degree of orthogonality, while the latter limits the expressive power of neural networks and leaves the orthogonality (expressive power) unchangeable. Our method COCK overcomes these weakness by adjusting the restriction to the parameter space. The work most similar to ours is layer-wise orthogonal training (LOT) (Xu et al., 2022). Both of us utilize the ONI method to reparameterize convolution kernels. However, the learnable parameters of ours are in the frequency domain, while LOT learns parameters in the spatial domain. More importantly, COCK is able to adjust the expressive power by altering its two hyperparameters. The hyperparameters provide a broader range of expressive power and enable COCK to be a tool to investigate the trade-off between expressive power and generalization.

**Relation Between Vanilla Classification Accuracy and Adversarial Robustness.** Adversarial training (Madry, 2017) and its certified variants (Wong & Kolter (2018); Huang et al. (2021); Zhang et al. (2021); Gowal et al. (2018)) which involved minimizing a worst-case loss (or its approximation) using uniformly-bounded perturbations to the training data are the most well-known approaches in adversarial defenses. These approaches imply the existence of a discrepancy between vanilla classification accuracy and adversarial robustness. Upon this discrepancy, several regularization-based methods (Raghunathan et al. (2020); Leino et al. (2021); Zhang et al. (2019); Hoffman et al. (2019); Gouk et al. (2020); Zhang et al. (2019)) have been proposed to make a trade-off between vanilla classification accuracy and adversarial robustness. Typically, Zhang et al. (2019) decomposed robust error as the sum of classification error and boundary error and further undertook the trade-off by introducing a weight factor. Our paper explores the discrepancy between vanilla classification accuracy and certified robustness standing on the trade-off between expressive power and generalization. We find an empirical relation between the discrepancy (between vanilla classification accuracy and certified robustness) and the trade-off (between expressive power and generalization).

## 3 PRELIMINARIES

We begin by introducing the basic notations and common practice of the fundamental problem framework in machine learning, certified robustness in Lipschitz-constrained neural networks, and orthogonal convolutions.

### 3.1 CHARACTERIZATIONS OF RISK AND ACCURACY

Consider a true data distribution $p(\boldsymbol{x}, \boldsymbol{y})$ and the sampled training sets $\mathbb{D} \sim p(\boldsymbol{x}, \boldsymbol{y})$ of size $N$. Represent the neural network as $F_\theta(\cdot)$ and the training set as $\{(\boldsymbol{x}^{(i)}, \boldsymbol{y}^{(i)})\}_{i=1}^N$. Training the model can be viewed as tuning the parameters to minimize the discrepancy between the desired output $\boldsymbol{y}$ and the predicted output $F_\theta(\boldsymbol{x})$. The empirical risk averaged over the sample loss $l(\boldsymbol{y}, F_\theta(\boldsymbol{x}))$ is defined as:

$$\mathcal{L}_\theta = \frac{1}{N} \sum_{i=1}^N l(\boldsymbol{y}^{(i)}, F_\theta(\boldsymbol{x}^{(i)})). \tag{1}$$

The expected risk under the true data distribution is defined as:

$$\mathcal{L}_\theta^* = \mathbb{E}_{(\boldsymbol{x}, \boldsymbol{y}) \sim p(\boldsymbol{x}, \boldsymbol{y})}[l(\boldsymbol{y}, F_\theta(\boldsymbol{x})]. \tag{2}$$

Consider a classification problem with one-hot encoding. Let $t_x$ be the ground-truth class of $\boldsymbol{x}$. The vanilla classification accuracy on the training set is defined as

$$Acc = \frac{1}{N} \sum_{i=1}^N \mathbb{I}\{t_{x^{(i)}} = \operatorname{argmax}_j F_\theta(\boldsymbol{x}^{(i)})_j\}, \tag{3}$$

where $\mathbb{I}\{\cdot\}$ is the indicator function. The vanilla classification accuracy on the test set is defined as:

$$Acc^* = \mathbb{Pr}\{\mathbb{I}\{t_x = \operatorname{argmax}_j F_\theta(\boldsymbol{x})_j\}\}. \tag{4}$$

It is long believed that weight regularization helps alleviate overfitting and improve generalization. A general idea of weight regularization is to provide layer-wise constraints on the weights during optimization, which can be formulated as

$$\theta^* = \operatorname{argmin}_\theta \mathcal{L}_\theta$$
$$s.t. \quad \gamma(\theta) \tag{5}$$

where $\gamma(\theta)$ are the layer-wise constraints imposed on the weight parameters.

## 3.2 DESCRIPTIONS OF CERTIFIED ROBUSTNESS IN LIPSCHITZ-CONSTRAINED MODELS

**Definition 3.2.1.** (Vershynin, 2018) *Let $(\mathbb{X}, d_{\mathbb{X}})$ and $(\mathbb{Y}, d_{\mathbb{Y}})$ be metric spaces. A function $f : \mathbb{X} \to \mathbb{Y}$ is called Lipschitz if there exists $L \in \mathbb{R}$ such that*

$$d_{\mathbb{Y}}(f(u), f(v)) \leq L d_{\mathbb{X}}(u, v) \; for \; every \; u, v \in \mathbb{X}. \tag{6}$$

*The infimum of all $L$ in this definition is called the Lipschitz norm of $f$ and is denoted $||f||_{Lip}$.*

For Lipschitz-constrained neural networks, Tsuzuku et al. (2018) provided the certified robustness when both $d_{\mathbb{X}}$ and $d_{\mathbb{Y}}$ are defined by $l_2$ norm. Let $L_F$ be the Lipschitz norm of network $F$ when $d_{\mathbb{X}}$ and $d_{\mathbb{Y}}$ are both in $l_2$ norm. Denote the minimum output margin

$$M_{F,x} = F(x)_{t_x} - \max_{i \neq t_x}\{F(x)_i\}. \tag{7}$$

The following theorem holds.

**Theorem 3.2.2.** (Tsuzuku et al., 2018) *If $\sqrt{2}L_F||\epsilon||_2 \leq M_{F,x}$, then $M_{F,x+\epsilon} \geq 0$. That is, the network $F$ is certified robust in $x$.*

By imposing layer-wise weight regularization to regularize the spectrum, we can achieve a Lipschitz-constrained neural network. We consider a linear layer with input $x \in \mathbb{R}^{d_{in} \times 1}$, weight $W \in \mathbb{R}^{d_{out} \times d_{in}}$ and output $y \in \mathbb{R}^{d_{out} \times 1}$

$$y = Wx. \tag{8}$$

Typically, by leveraging layer-wise orthogonal constraints[3], the Lipschitz norm of the linear layer is 1. Note that most of the activations are contractive (Szegedy et al., 2013), the Lipschitz norm of the whole network with layer-wise orthogonal constraints is strictly restricted to 1.

## 3.3 EXPRESSIONS OF ORTHOGONAL CONVOLUTION

We can easily extend the layer-wise orthogonal weight regularization to convolutions. A convolution layer is parameterized by weights $W \in \mathbb{R}^{d_{in} \times d_{out} \times F_h \times F_w}$, where $F_h$ and $F_w$ are the height and width of the filter. Take feature maps (activations) $X \in \mathbb{R}^{d_{in} \times h \times w}$ as input and denote the convolution operation as $Y = Conv(W, X)$.

**Definition 3.3.1.** *Represent the convolution operation $Y = Conv(W, X)$ as linear transform $vec(Y) = M vec(X)$, where $vec(\cdot)$ flattens the height $h$ and width $w$ into $hw$. The convolution kernel is orthogonal if and only if the singular values of the corresponding Jacobian matrix $M$ are all 1. Further, Convolution Neural Networks are called Orthogonal Convolution Neural Networks if all the convolution kernels are orthogonal.*

**Remark 3.3.1.1.** When the Jacobian matrix $M$ is not a square matrix, we generalize the definition of orthogonal convolution kernel from all singular values to $d$ singular values due to the rank constraint, where $d$ is the minimum of the two dimensions of $M$.

## 4 CAPTURE THE TRADE-OFF FOR CERTIFIED ROBUSTNESS

We firstly establish an equivalence between the convergence of training loss and the improvement of certified robustness in Section 4.1. This proposition suggests that there is a trade-off between expressive power and generalization for certified robustness. Based on this equivalence, we define the risk and accuracy for certified robustness in Section 4.2.

---

[3]If $d_{in} \geq d_{out}$, we regularize weights as $WW^T = I_{d_{out}}$. Otherwise, we regularize weights as $W^T W = I_{d_{in}}$. $I_d$ is the identity with dimension $d$.

## 4.1 EQUIVALENCE BETWEEN LOSS CONVERGENCE AND CERTIFIED ROBUSTNESS

Under the setting in Section 3.1, the empirical risk averaged over the sample loss defined by Cross-Entropy (CE) loss is defined as:

$$\mathcal{L}_\theta = \frac{1}{N} \sum_{i=1}^{N} l_{CE}(\boldsymbol{y}^{(i)}, F_\theta(\boldsymbol{x}^{(i)})). \tag{9}$$

For simplicity, we consider a $d$-class classification problem with one-hot encoding:

$$\mathcal{L}_\theta = \frac{1}{N} \sum_{i=1}^{d} \sum_{j=1}^{c_i} -log \frac{e^{F_\theta(\boldsymbol{x}^{(ij)})_i}}{\sum_{k=1}^{d} e^{F_\theta(\boldsymbol{x}^{(ij)})_k}}, \tag{10}$$

where $\boldsymbol{x}^{(ij)}$ is the $j$-th sample of class $i$ and $c_i$ is the amount of samples of class $i$. As usual, we add a $softmax(\cdot)$ after the output of the neural network. The following theorem holds.

**Theorem 4.1.1.** *For a Lipschitz-constrained neural network adopting classification loss* (10) *during training, the decrease of training loss is equivalent to the improvement of certified robustness when* Theorem 3.2.2 *holds.*

*Proof.* For a sample point $\boldsymbol{x}^{(ij)}$ of the $i$-th class, define $m_k^{(ij)} = F_\theta(\boldsymbol{x}^{(ij)})_i - F_\theta(\boldsymbol{x}^{(ij)})_k$ as the output margin of $k$ for $i$-th class. Hence,

$$\mathcal{L}_\theta = \frac{1}{N} \sum_{i=1}^{d} \sum_{j=1}^{c_i} log(1 + \sum_{k!=i} e^{-m_k^{(ij)}}). \tag{11}$$

Trivially, minimizing training loss is equivalent to maximizing the output margin $m_k^{(ij)}$.

Note that $L_F$ holds during training because the network is Lipschitz-constrained, by Theorem 3.2.2 we therefore know that certified robustness improves when training loss decreases. $\square$

If a Lipschitz-constrained network is powerful enough to cover the training set well, it will be extremely certified robust on the training set. However, excessive expressive power brings high generalization error to classification performance during inference (overfitting) and is certain to degrade the certified robustness on the test set. Therefore, there is a trade-off between expressive power and generalization for certified robustness, similar to the vanilla classification accuracy.

## 4.2 DEFINITION OF RISK AND ACCURACY FOR CERTIFIED ROBUSTNESS

According to the trade-off between expressive power and generalization, we can define the risk and accuracy for certified robustness for Lipschitz-constrained models corresponding to descriptions in machine learning. To begin with, we define the margin loss[4].

**Definition 4.2.1.** **(Margin Loss)** *Utilizing the definition of output margin in* (7)*, the margin loss is any loss whose reduction leads to the enlargement of output margin.*

**Remark 4.2.1.1.** The CE loss is a special kind of margin loss, which plays a crucial role in the equivalence above.

Typically, we can define the margin loss for the perturbation $||\epsilon||_2$ within $l_2$ norm.

**Definition 4.2.2.** **($\epsilon_2$-Margin Loss)** *Utilizing the definition of output margin in* (7)*, the $\epsilon_2$-margin loss is any loss whose reduction leads to the enlargement of those output margins that less than $\epsilon$ in the sense of $l_2$ norm.*

The definition of empirical ($\epsilon_2$-)certified robust risk and expected ($\epsilon_2$-)certified robust risk can be generalized from the traditional empirical risk and expected risk by simply replacing loss $l$ with ($\epsilon_2$-)margin loss.

Next, by transforming the definition in (3;4) using the output margin, we define the certified robust accuracy under the setting in Section 3.1.

---

[4]For Lipschitz-constrained models, the output margin becomes the only factor affecting the certified robustness.

**Definition 4.2.3. (Empirical Certified Robust Accuracy)**

$$ECR = \frac{1}{N} \sum_{i=1}^{N} (\mathbb{I}\{M_{F_\theta, x^{(i)}} \geq \sqrt{2} L_F \|\epsilon\|_2\}). \tag{12}$$

**Definition 4.2.4. (Expected Certified Robust Accuracy)**

$$ECR^* = \Pr\{M_{F_\theta, x} \geq \sqrt{2} L_F \|\epsilon\|_2\}. \tag{13}$$

## 5 PROPOSE OF COCK: CONTROLLABLE ORTHOGONAL CONVOLUTION KERNEL

To investigate the trade-off discussed in Section 4.1, we consider the setting of CNNs trained on real-world datasets such as CIFAR and ImageNet, for the reason that the certified robustness on MNIST is satisfactory enough. Though the orthogonal convolution regularizes the parameter space for the 1-Lipschitz property, it still suffers from the lack of strength and flexibility in expressive power. To overcome its weakness in expressive power while still exploiting its advantages in Lipschitz norm, we design a more powerful orthogonal convolution-Controllable Orthogonal Convolution Kernel (COCK). We first present the details of COCK in Section 5.1 and then prove that COCK is guaranteed to be more certified robust than other general orthogonal convolutions in Section 5.2. The improvement of certified robustness owing to the enhancement of expressive power aligns well with our understanding in Section 4.1.

### 5.1 DESIGN OF COCK

As usual, we take $h = w = n$ and $F_h = F_w = s$ as the kernel size. We characterize the singular values of the Jacobian matrix leveraging the conclusion in Sedghi et al. (2018). Let $\sigma(\cdot)$ represent the singular values.

**Proposition 5.1.1.** (Sedghi et al., 2018) *For any given convolution* $\mathbf{W} \in \mathbb{R}^{d_l \times d_{l-1} \times F_h \times F_w}$, $\forall u, v \in [F_h] \times [F_w]$, *undertake Fourier Transform* $\boldsymbol{P}^{(u,v)}[c,d] = (\boldsymbol{F}^T \mathbf{W}[c,d,:,:] \boldsymbol{F})[u,v], (c,d) \in [d_{l-1}] \times [d_l]$, *where* $\boldsymbol{F} \in \mathbb{R}^{s \times s}$ *is the corresponding DFT matrix, it holds* $\sigma(M) = \bigcup_{(u,v) \in [s] \times [s]} \sigma(\boldsymbol{P}^{(u,v)})$.

The proposition 5.1.1 suggests that the singular values of Jacobian matrix $\boldsymbol{M}$ are the union of the singular values of $s^2$ kernel matrices $\boldsymbol{P}^{(u,v)}$. We intend to orthogonalize the convolution layer via orthogonalizing each kernel, leveraging ONI algorithm (Huang et al., 2020) for reparameterization. We present the Algorithm of COCK in Appendix A.1. We can directly plug the COCK in CNNs, where the proxy parameter in Algorithm 1 is the learnable parameter which is in the Fourier frequency domain.

Generally, the Newton Iteration steps $t$ controls the orthogonality of the convolution layer. As $t$ increases, the orthogonality will be more strict. To further control the expressive power of COCK, we share the parameters of some convolution kernels by handcrafting the amount of distinct convolution kernel $k$. To be specific, given $k$, the $s^2$ kernels are formed by $k$ different kernels and the remaining $s^2 - k$ kernels reuse the parameters of the $k$ kernels. In this sense, the effective parameters of the convolution layer are actually $\boldsymbol{W} \in \mathbb{R}^{d_l \times d_{l-1} \times k}$. Intuitively, the increase of $k$ represents the improvement of expressive power. Intrinsically, this trick can be viewed as the regularization on parameter space as well, by sharing parameters. Overall, we can adjust the expressive power of COCK by altering these two parameters.

#### 5.1.1 ON THE EXPRESSIVE POWER OF COCK

The trade-off between expressive power and orthogonality intuitively makes COCK more powerful than general orthogonal convolutions. To illustrate this property, we consider the case $k = s^2$. Intuitively, from the perspective of parameter space, by adjusting proxy parameters, we can obtain countless kinds of singular values distribution, restricting all singular values $\in [0, 1]$. In particular, when $t \to \infty$, all singular values converge to 1. However, for general orthogonal convolutions, all singular values are strictly restricted to 1. From the perspective of data transformation, orthogonal

convolution is geometrically equivalent to a rotation or symmetry, while COCK also provides data scaling.

We undertake a toy example in two-dimension space. Consider the linearly inseparable 2-class classification problem and adopt a simple neural network with a linear layer and an activation as the classifier. We compare the orthogonal neural network whose linear layer is orthogonal with another network whose linear layer is a diagonal matrix with eigenvalues $\in [0,1]^5$. Both the two neural networks adopt the same activation r-ReLU defined as follows.

**Definition 5.1.1.1. (r-ReLU Activations)**

$$\text{r-ReLU}(\boldsymbol{x}) = \text{ReLU}(||\boldsymbol{x}||_2 - r). \tag{14}$$

We visualize the classification result in Figure 1. The linearly inseparable problem is to classify the four data points in two classes. The four points initially lie around the biggest circle (black) with radius $R > r$. For the first neural network equipped with the orthogonal linear layer, after passing through the linear layer, all points still lie around the black circle. Then, after passing through the activation, the outputs of all points are the same, which fails to classify the problem. For the second neural network on the contrary, by carefully learning (updating) the eigenvalues of the linear layer, the points can distribute around an ellipse (red) whose long axis is greater than $r$, while its minor axis is shorter than $r$. At last, after passing through the activation, the outputs of the four points differ according to their class.

Figure 1: A toy example—the XOR binary classification problem. The problem is to classify the four points lie around the biggest circle (black). For standard orthogonal convolutions, the four points still lie around the biggest circle while COCK is able to transform the four points to the red ellipse and solve the XOR problem.

## 5.2 On the Certified Robustness of COCK

According to our understanding in Section 4.1 that enough power to cover the dataset is equivalent to the improvement of certified robustness, we in this section present the proposition that COCK is more certified robust than general orthogonal convolutions owing to its improvement of expressive power.

**Theorem 5.2.1.** *Assuming a well-conditioned optimization, the certified robustness of COCK is not inferior to general orthogonal convolutions.*

The proof is shown in Appendix A.2. Under the assumption of the well-conditioned optimization, we undertake the proof from the view of parameter existence. The theoretical result aligns well with our understanding of the equivalence between the convergence of training loss and the improvement of certified robustness.

**Remark 5.2.1.1.** Up to now, our formulation of certified robustness merely considers perturbations and Lipschitz norm with respect to $l_2$ norm. We generalize to the case that $d_{\mathbb{X}}$ is in $l_p$ norm and $d_{\mathbb{Y}}$ is in $l_q$ norm. Correspondingly, the perturbation $||\epsilon||$ is within $l_p$ norm $||\epsilon||_p$. Details are discussed in Appendix A.3.

## 6 Experiment

In this section, we intend to answer and verify the following questions empirically:

- Does the improvement of expressive power enable COCK to cover the training set better and further improve the certified robustness on the training set? Will this improvement make the model more certified robust during inference?

---

[5] COCK provides the eigenvalues of linear layer $\in [0,1]$. Hence, our design of the diagonal matrix is a special case of COCK.

- By altering the power of the whole neural network via COCK, where is the optimal trade-off between expressive power and generalization? To be specific, will the optimal trade-off for certified robustness keep pace with the optimal one for vanilla classification accuracy during inference?

## 6.1 SETUP

**Datasets and Models.** We try out different amounts of layers (blocks) of LipConvNet (Anil et al. (2019), Behrmann et al. (2019)) including LipConvNet-5, LipConvNet-15 and LipConvNet-35 on CIFAR-10 (Krizhevsky, 2009), CIFAF-100 (Krizhevsky, 2009) and TinyImageNet (Russakovsky et al., 2015). The activation function we use is max-min activation.

**Baseline Methods.** To answer the first question, we compare COCK with existing orthogonal convolutions including SOC(Singla & Feizi, 2021), BCOP(Li et al., 2019) and Cayley(Trockman & Kolter, 2021). All of those methods are adopted in LipConvNets.

**Hyperparameters.** To investigate the second question, we alter the expressive power of COCK by two hyperparameters in COCK, the amount of distinct convolution kernel $k$ and the ONI steps $t$. Generally, the increase of $k$ corresponds to the improvement of expressive power and $t$ controls the trade-off between expressive power and orthogonality, or generalization.

**Training Loss and Certified Robust Accuracy.** Based on Theorem 4.1.1, we adopt vanilla CE loss during training rather than CE loss with any regularization on output margin. We utilize the Definition 4.2.4 to evaluate certified robust accuracy within $l_2$ norm perturbation $\epsilon = \frac{36}{255}, \epsilon = \frac{72}{255}$ and $\epsilon = \frac{108}{255}$.

Our experiment is organized into two parts and each for one question above. In Section 6.2, we mainly verify if COCK is empowered to cover the training set better than baseline methods. In Section 6.3, we empirically investigate the trade-off between expressive power and generalization for certified robustness. We undertake an ablation study on ONI steps $t^6$ in Appendix A.4.1.

## 6.2 ENOUGH POWER TO COVER THE TRAINING SET IMPROVES CERTIFIED ROBUSTNESS

Results on CIFAR-10 are shown in Table 1 and results on CIFAR-100 and TinyImageNet are listed respectively in Appendix A.4.4 and Appendix A.4.5. We additionally present the training loss correspondingly in Appendix A.4.2 to visualize the convergence on the training set.

Remarkably, owing to the improvement of expressive power, COCK covers the training set much better than other orthogonal convolutions, resulting in the improvement both in vanilla classification accuracy and certified robustness. Additionally, the improvement of certified robustness on the training set generalizes to the test set to some extent, in particular when attack perturbation is large.

It is worth mentioning that results in Table 1, 4, 5 are tuned for a better test certified robust accuracy by altering the hyperparameters of COCK, resulting from the trade-off between expressive power and generalization for certified robustness. Similarly, by controlling the trade-off via adjusting the hyperparameters, we can also acquire a better vanilla test accuracy comparable to the best performance of the baseline orthogonal convolutions. Results are shown in Appendix A.4.6-A.4.8. The discrepancy between vanilla classification accuracy and certified robustness in the optimal trade-off between expressive power and generalization will be further discussed in Section 6.3.

Likewise, controlling the trade-off by the hyperparameters, COCK nearly fully covers the training set. We take LipConvNet-15 as an example and show the results when the amount of distinct convolution kernel $k = 5$ in Appendix A.4.3.

## 6.3 POWER-DRIVEN SHIFT FROM VANILLA CLASSIFICATION ACCURACY TO CERTIFIED ROBUSTNESS

To better understand the impact of the trade-off between expressive power and generalization for certified robustness, we empirically investigate the optimal trade-off for certified robustness in this

---

[6]The impact of the amount of distinct convolution kernel $k$ is trivial and can be seen in Section 6.3.

Table 1: Vanilla Classification Accuracy and Certified Robust Accuracy on CIFAR-10

| Stage | Model | Method | Accuracy(%) | $\epsilon = \frac{36}{255}$ | $\epsilon = \frac{72}{255}$ | $\epsilon = \frac{108}{255}$ |
|---|---|---|---|---|---|---|
| Training | LipConvNet-5 | SOC | 78.11 | 60.41 | 41.11 | 24.34 |
| | | BCOP | 84.51 | 67.02 | 45.51 | 26.19 |
| | | Cayley | 72.17 | 52.92 | 33.65 | 18.50 |
| | | COCK | **85.85** | **69.25** | **48.63** | **29.70** |
| | LipConvNet-15 | SOC | 80.19 | 63.30 | 44.06 | 27.45 |
| | | BCOP | 81.63 | 63.92 | 43.24 | 25.27 |
| | | Cayley | 77.25 | 59.54 | 40.06 | 23.77 |
| | | COCK | **90.09** | **85.90** | **80.13** | **73.35** |
| | LipConvNet-35 | SOC | 77.61 | 60.00 | 40.97 | 24.90 |
| | | BCOP | 63.52 | 43.76 | 25.59 | 12.33 |
| | | Cayley | 69.11 | 49.87 | 31.46 | 17.21 |
| | | COCK | **88.36** | **79.26** | **67.47** | **54.21** |
| Test | LipConvNet-5 | SOC | **75.60** | 59.61 | 42.28 | 27.09 |
| | | BCOP | 75.24 | 58.49 | 40.71 | 25.41 |
| | | Cayley | 71.62 | 54.37 | 36.55 | 21.66 |
| | | COCK | 75.12 | **59.82** | **43.98** | **30.03** |
| | LipConvNet-15 | SOC | **76.65** | 62.51 | 45.32 | 30.54 |
| | | BCOP | 74.31 | 58.12 | 40.21 | 25.70 |
| | | Cayley | 74.37 | 55.40 | 35.89 | 20.33 |
| | | COCK | 75.56 | **71.03** | **65.83** | **60.79** |
| | LipConvNet-35 | SOC | **73.85** | **58.55** | 42.14 | 27.36 |
| | | BCOP | 63.52 | 43.76 | 25.59 | 12.33 |
| | | Cayley | 68.04 | 50.82 | 33.67 | 19.96 |
| | | COCK | 67.56 | 57.97 | **47.35** | **37.48** |

section. We record the trend of test vanilla classification accuracy and test certified robust accuracy in Figure 2. Implementation details can be seen in Appendix A.4.9.

We notice that though overfitting and underfitting still exist for certified robustness, the optimal trade-off between expressive power and generalization for certified robustness is more powerful than the vanilla classification accuracy during inference on the test set and this phenomenon is more remarkable for larger attack perturbations. We call this phenomenon a power-driven shift. Intuitively, this can be partly attributed to the fact that models need to be more powerful to enlarge the output margin since certified robustness can be viewed as a more difficult problem than vanilla classification. This experimental result implies that by carefully improving the expressive power from the optimal trade-off for vanilla classification performance, the model can be more certified robust. We leave the theoretical establishment to explain this interesting phenomenon in future work.

# 7 CONCLUSION

Our paper mainly connects certified robustness with the fundamental machine learning framework for Lipschitz-constrained models. The equivalence between enough power to cover the training set and the improvement of certified robustness for Lipschitz-constrained models exposes that there is a trade-off between expressive power and generalization (assuming a well-conditioned optimization) for certified robustness. We provide key insight into understanding the gap between training and testing for certified robustness. Empirically, we observe that there is a power-driven shift from vanilla classification accuracy to certified robust accuracy in the sense of the optimal trade-off between expressive power and generalization. This phenomenon suggests that expressive power is crucial for certified robustness both on the training set and on the test set. By carefully improving the expressive power from the optimal trade-off for vanilla classification performance, we can obtain higher certified robustness.

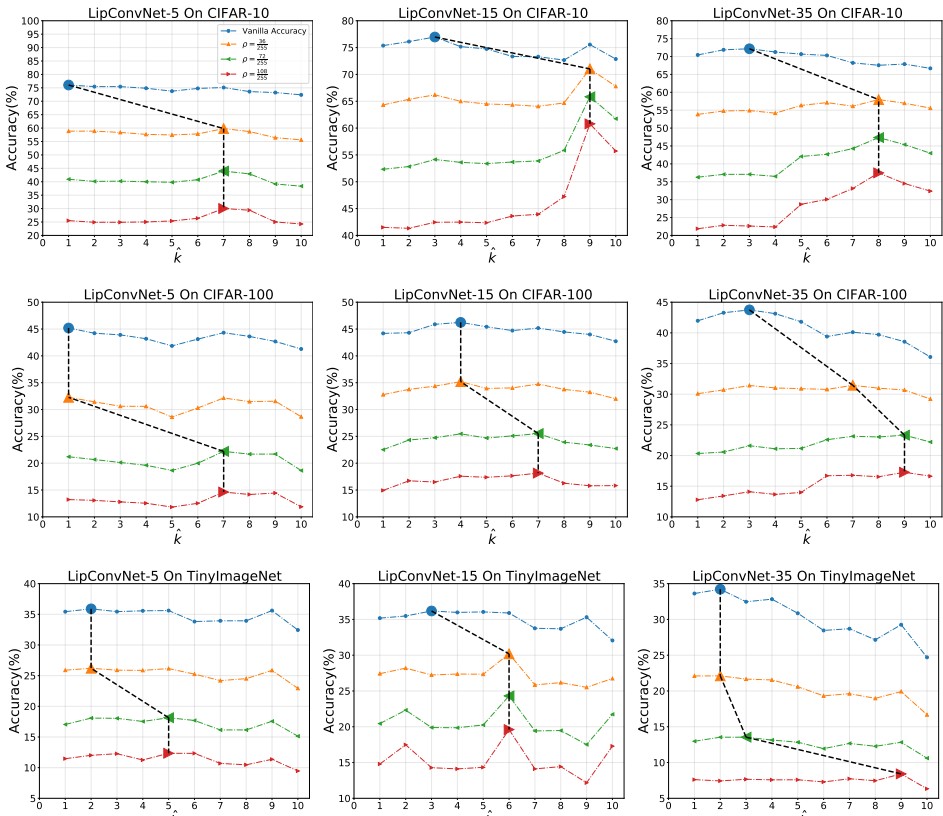

Figure 2: The Trade-off Between Expressive Power and Generalization. The optimal trade-off for vanilla classification accuracy and certified robustness is highlighted. The index $\hat{k} = (k_1, k_2, ..., k_L)$ indicates the combination of $k$ in a $L$-layer network. Typically, the larger $\hat{k}$ is, the more powerful the model will be. Note that the value of $\hat{k}$ in figures merely represents the strength rather than practical values and the same index in different figures does not represent the same combination.

## 7.1 LIMITATIONS AND FUTURE WORK

**The lack in consideration of optimization.** Empirically, though we have attempted plenty of optimization settings by adjusting the optimizer such as Adam and SGD and learning rate schedules to alleviate the impacts of optimization, there always a gap between training loss and expressive power induced by optimization. Theoretically, we always assumed a well-conditioned optimization and did not consider training dynamics in this paper.

**The insufficient precision in expressive power.** To empirically investigate the optimal trade-off between expressive power and generalization, we qualitatively adjusted the expressive power by altering the hyperparameters of COCK while we did not quantitatively describe the expressive power.

In the future, to close the gap between training and testing for certified robustness, we will provide some theoretical evidence on the power-driven shift, or to be general, the generalization behavior of certified robustness.

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

## A APPENDIX

### A.1 OVERALL ALGORITHM

We first recapture the ONI algorithm (Huang et al., 2020). To be specific, let $\boldsymbol{W} = (\boldsymbol{V}\boldsymbol{V}^T)^{-\frac{1}{2}}\boldsymbol{V}$, it holds

$$\boldsymbol{W}\boldsymbol{W}^T = (\boldsymbol{V}\boldsymbol{V}^T)^{-\frac{1}{2}}\boldsymbol{V}\boldsymbol{V}^T(\boldsymbol{V}\boldsymbol{V}^T)^{-\frac{1}{2}} = \boldsymbol{I}. \tag{15}$$

We utilize Newton Iteration to compute $(\boldsymbol{V}\boldsymbol{V}^T)^{-\frac{1}{2}}$.

**Theorem A.1.1.** (Huang et al., 2020) *Given matrix $\boldsymbol{V}$, let $\boldsymbol{S} = \boldsymbol{V}\boldsymbol{V}^T$. Undertake recurrence*

$$\boldsymbol{B}_0 = \boldsymbol{I}, \ \boldsymbol{B}_t = \frac{3}{2}\boldsymbol{B}_{t-1} - \frac{1}{2}\boldsymbol{B}_{t-1}^3\boldsymbol{S}. \tag{16}$$

*If spectral norm $||\boldsymbol{V}|| < 1$ then $\boldsymbol{B}_t \to \boldsymbol{S}^{-\frac{1}{2}}$, $t \to \infty$.*

The ONI algorithm reparameterizes orthogonal weight $W$ by the recurrence above. Leveraging the ONI algorithm, we present the overall algorithm of COCK in Algorithm 1. In Algorithm 1, the operation $\frac{\boldsymbol{Z}}{||\boldsymbol{Z}||_F}$ is to satisfy the convergence condition:

$$||\boldsymbol{V}|| < 1. \tag{17}$$

---

**Algorithm 1 COCK**

---

**Require:** Input $\mathbf{X} \in \mathbb{R}^{d_{l-1} \times n \times n}$, Proxy parameter $\mathbf{P} \in \mathbb{R}^{d_l \times d_{l-1} \times s \times s}$
**Ensure:** Output of convolution layer $\mathbf{Y}$
    **for** $u \leftarrow 1$ to $s$ **do**
      **for** $v \leftarrow 1$ to $s$ **do**
        $\boldsymbol{Z} \leftarrow \mathbf{P}[:,:,u,v] \in \mathbb{R}^{d_l \times d_{l-1}}$
        $\boldsymbol{V} \leftarrow \dfrac{\boldsymbol{Z}}{||\boldsymbol{Z}||_F}$
        $\boldsymbol{S} \leftarrow \boldsymbol{V}\boldsymbol{V}^T$
        $\boldsymbol{B}_0 \leftarrow \boldsymbol{I}$
        **for** $\tau \leftarrow 1$ to $t$ **do**
          $\boldsymbol{B}_\tau \leftarrow \dfrac{3}{2}\boldsymbol{B}_{\tau-1} - \dfrac{1}{2}\boldsymbol{B}_{\tau-1}^3\boldsymbol{S}$            ▷ Undertake ONI for each kernel
        **end for**
        $\mathbf{W}[:,:,u,v] \leftarrow \boldsymbol{B}_t\boldsymbol{V}$
      **end for**
    **end for**
    **for** $i \leftarrow 1$ to $d_l$ **do**
      **for** $j \leftarrow 1$ to $d_{l-1}$ **do**
        $\boldsymbol{W}[i,j,:,:] \leftarrow IFFT_{2D}(\boldsymbol{W}[i,j,:,:])$         ▷ Transform to the spatial domain
      **end for**
    **end for**
    $\mathbf{Y} \leftarrow Conv(\mathbf{W}, \mathbf{X})$
    **return** $\mathbf{Y}$

---

### A.2 OMITTED PROOFS

#### A.2.1 USEFUL LEMMAS

We begin by a basic lemma for ONI:

**Lemma A.2.1.1.** (Guo & Higham, 2006) *For Newton Iteration:*

$$\boldsymbol{X}_0 = \boldsymbol{I}, \ \boldsymbol{X}_{t+1} = \frac{1}{2}[3\boldsymbol{X}_t - \boldsymbol{X}_t^3\boldsymbol{S}]. \tag{18}$$

*The residual term $\boldsymbol{R}_t = \boldsymbol{I} - \boldsymbol{X}_t^2 \boldsymbol{S}$ satisfies*

$$\boldsymbol{R}_{t+1} = \frac{3}{4}\boldsymbol{R}_t^2 + \frac{1}{4}\boldsymbol{R}_t^3. \tag{19}$$

*If $||\boldsymbol{R}_0|| < 1$, then $\{||\boldsymbol{R}_t||\}$ monotonically decreases to zero.*

**Lemma A.2.1.2.** *If $\boldsymbol{A}, \boldsymbol{B}, \boldsymbol{AB} \in \mathbb{S}_{++}^n$, that is, they are symmetric positive matrices, then $\boldsymbol{AB} = \boldsymbol{BA}$.*

*Proof.*

$$(\boldsymbol{AB})^T = \boldsymbol{B}^T \boldsymbol{A}^T = \boldsymbol{BA}. \tag{20}$$

Note that

$$(\boldsymbol{AB})^T = \boldsymbol{AB}, \tag{21}$$

It holds

$$\boldsymbol{AB} = \boldsymbol{BA}. \tag{22}$$

$\square$

**Lemma A.2.1.3.** *If $\boldsymbol{A}, \boldsymbol{B}, \boldsymbol{AB} \in \mathbb{S}_{++}^n$, then $\boldsymbol{A}^{\frac{1}{2}} \boldsymbol{B}^{\frac{1}{2}} = \boldsymbol{B}^{\frac{1}{2}} \boldsymbol{A}^{\frac{1}{2}}$.*

*Proof.* Take one of the eigenvectors of $\boldsymbol{A}^{\frac{1}{2}}$ as $\boldsymbol{x}$ and denote $\lambda$ as the corresponding eigenvalue. It holds

$$\boldsymbol{A}^{\frac{1}{2}} \boldsymbol{x} = \lambda \boldsymbol{x}. \tag{23}$$

Further

$$\boldsymbol{A}\boldsymbol{x} = \boldsymbol{A}^{\frac{1}{2}} \boldsymbol{A}^{\frac{1}{2}} \boldsymbol{x} = \boldsymbol{A}^{\frac{1}{2}} \lambda \boldsymbol{x} = \lambda \boldsymbol{A}^{\frac{1}{2}} \boldsymbol{x} = \lambda^2 \boldsymbol{x}. \tag{24}$$

That means $\boldsymbol{x}$ is one of the eigenvectors of $\boldsymbol{A}$. In turn, for any eigenvalue $\lambda$ of $\boldsymbol{A}$, the corresponding eigenvalue of $\boldsymbol{A}^{\frac{1}{2}}$ is $\lambda^{\frac{1}{2}}$. Note that

$$\begin{aligned} \boldsymbol{A}\boldsymbol{x} &= \lambda x, \\ \boldsymbol{A}^{\frac{1}{2}} \boldsymbol{y} &= \lambda^{\frac{1}{2}} \boldsymbol{y}, \\ \boldsymbol{A}^{\frac{1}{2}} \boldsymbol{A}^{\frac{1}{2}} \boldsymbol{y} &= \lambda \boldsymbol{y}, \\ \boldsymbol{A}\boldsymbol{y} &= \lambda \boldsymbol{y}. \end{aligned} \tag{25}$$

Hence, $\boldsymbol{x}$ and $\boldsymbol{y}$ belong to the same eigen-subspace. Further, the eigenvectors of $\boldsymbol{A}$ and $\boldsymbol{A}^{\frac{1}{2}}$ are same. Likewise, the eigenvectors of $\boldsymbol{B}$ and $\boldsymbol{B}^{\frac{1}{2}}$ are same too.

Utilizing lemma A.2.1.2 we obtain that $\boldsymbol{A}, \boldsymbol{B}$ are commutative. Hence,

$$\begin{aligned} \boldsymbol{A}, \boldsymbol{B} \text{ are commutative} &\iff \boldsymbol{A}, \boldsymbol{B} \text{ have same eigenvectors} \\ &\iff \boldsymbol{A}^{\frac{1}{2}}, \boldsymbol{B}^{\frac{1}{2}} \text{ have same eigenvectors} \\ &\iff \boldsymbol{A}^{\frac{1}{2}}, \boldsymbol{B}^{\frac{1}{2}} \text{ are commutative.} \end{aligned} \tag{26}$$

$\square$

### A.2.2 COCK Is 1-LIPSCHITZ

We utilize Theorem 3.2.2 to characterize the certified robustness. The following theorem suggests that COCK is Lipschitz-constrained with Lipschitz norm less than one. Note that the convergence condition requires $||\boldsymbol{I} - \boldsymbol{S}|| < 1$. Therefore, in convolution, we assume that all the singular values of $\boldsymbol{S}$ are positive. Actually, the zero singular values of $\boldsymbol{S}$ do not contribute expressive power which do not affect the training of the network.

**Theorem A.2.2.1.** *Assuming that the Newton Iteration steps is $t$, then the singular values of the output matrix from ONI algorithm are restricted to $[\sqrt{1 - \lambda_{max}(\boldsymbol{R}_t)}, \sqrt{1 - \lambda_{min}(\boldsymbol{R}_t)}]$.*

*Proof.* Firstly, we are going to prove that

$$\boldsymbol{S}^{-1}, \boldsymbol{I} - \boldsymbol{R}_k, (\boldsymbol{I} - \boldsymbol{R}_k)\boldsymbol{S}^{-1} \in \mathbb{S}_{++}^n. \tag{27}$$

On one hand, by $\boldsymbol{V} = \dfrac{\boldsymbol{Z}}{||\boldsymbol{Z}||_F}$, the singular values of $\boldsymbol{S} \in [0,1]$. On the other hand, by our assumption all singular values of $\boldsymbol{S}$ are positive, then $\boldsymbol{S}, \boldsymbol{S}^{-1} \in \mathbb{S}_{++}^n$.

On one hand, from $\boldsymbol{R}_0 = \boldsymbol{I} - \boldsymbol{S}$ we know that singular values of $\boldsymbol{R}_0 \in [0,1)$. On the other hand, utilizing lemma A.2.1.1 we obtain

$$\boldsymbol{R}_{k+1} = \frac{3}{4}\boldsymbol{R}_k^2 + \frac{1}{4}\boldsymbol{R}_k^3, \sigma(\boldsymbol{R}_k) \in [0,1). \tag{28}$$

Hence, $\boldsymbol{I} - \boldsymbol{R}_k \in \mathbb{S}_{++}^n$ and $0 < \sigma(\boldsymbol{I} - \boldsymbol{R}_k) \le 1$.

Note that

$$\boldsymbol{X}_0 = \boldsymbol{I}, \ \boldsymbol{X}_{k+1} = \frac{1}{2}[3\boldsymbol{X}_k - \boldsymbol{X}_k^3\boldsymbol{S}], \tag{29}$$

where $\boldsymbol{X}_k = \text{polynomial}(\boldsymbol{S})$, therefore, $\boldsymbol{X}_k$ is symmetric owing to $\boldsymbol{S}$ is symmetric. Further, $\boldsymbol{X}_k^2$ is symmetric. Hence, $(\boldsymbol{I} - \boldsymbol{R}_k)\boldsymbol{S}^{-1} \in \mathbb{S}_{++}^n$. Here we finish the proof of (27).

Finally, leverage lemma A.2.1.3 we have

$$(\boldsymbol{I} - \boldsymbol{R}_k)^{\frac{1}{2}}\boldsymbol{S}^{-\frac{1}{2}} = \boldsymbol{S}^{-\frac{1}{2}}(\boldsymbol{I} - \boldsymbol{R}_k)^{\frac{1}{2}}. \tag{30}$$

Hence,

$$\begin{aligned} \boldsymbol{W}\boldsymbol{W}^T &= [(\boldsymbol{I} - \boldsymbol{R}_k)\boldsymbol{S}^{-1}]^{\frac{1}{2}}\boldsymbol{V}\boldsymbol{V}^T[(\boldsymbol{S}^{-1})^T(\boldsymbol{I} - \boldsymbol{R}_k)^T]^{\frac{1}{2}} \\ &= \boldsymbol{I} - \boldsymbol{R}_k. \end{aligned} \tag{31}$$

Therefore, we conclude that

$$\begin{aligned} \sigma_{max}(W) &= \sqrt{1 - \lambda_{min}(\boldsymbol{R}_k)}. \\ \sigma_{min}(W) &= \sqrt{1 - \lambda_{max}(\boldsymbol{R}_k)}. \end{aligned} \tag{32}$$

$\square$

For COCK whose convolution layer is concatenated by several outputs of ONI, the singular values of convolution layer is the union of those singular values. Hence, COCK is a Lipschitz-constrained with Lipschitz norm less than one.

### A.2.3 PROOF OF THEOREM 5.2.1

*Proof.* For a network represented as $F$, its certified robust accuracy can be expressed as

$$\Pr\{\sqrt{2}L_F||\epsilon||_2 \le M_{F,x}\} = \Pr\{\sqrt{2}L_F||\epsilon||_2 \le F(x)_{t_x} - \max_{i \ne t_x}\{F(x)_i\}\}. \tag{33}$$

For proxy parameter $\boldsymbol{Z} \in \mathbb{R}^{d_l \times d_{l-1}}$ in Algorithm 1, without loss of generality, we assume $d_l \ge d_{l-1} \triangleq d$. Further, let all singular values of $\boldsymbol{Z}$ be the same. Hence, by operation $\boldsymbol{V} = \dfrac{\boldsymbol{Z}}{||\boldsymbol{Z}||_F}$, the singular values of $\boldsymbol{S} = \boldsymbol{V}\boldsymbol{V}^T$ are all $\frac{1}{d}$. For simplicity, we can directly neglect the zero singular values of $\boldsymbol{S}$ and let $\boldsymbol{S} = \dfrac{1}{d}\boldsymbol{I}$ [7]. Further, by

$$\boldsymbol{R}_0 = \boldsymbol{I} - \boldsymbol{S} = (1 - \frac{1}{d})\boldsymbol{I}, \ \boldsymbol{R}_{t+1} = \frac{3}{4}\boldsymbol{R}_t^2 + \frac{1}{4}\boldsymbol{R}_t^3, \tag{34}$$

all singular values of $\boldsymbol{R}_t$ keep the same, denote $\sigma$. For a $L$-layer neural network, the Lipschitz norm $L_F^{COCK} \le \sigma^L$. While the Lipschitz norm of general orthogonal convolutions is $L_F^{ORTH} = 1$.

---

[7]For one thing, the zero singular values will not affect the compuatation of residual term $\boldsymbol{R}_k$ and the Lipschitz norm of the network depends on the maximum singular value. For another, the zero singular values do not contribute expressive power.

Consider a single convolution layer, for a standard orthogonal convolution layer equipped with any parameters, there exists certain parameters for COCK such that

$$F_{conv}^{COCK}(x)_{t_x} - \max_{i \neq t_x}\{F_{conv}^{COCK}(x)_i\} = \sigma(F_{conv}^{ORTH}(x)_{t_x} - \max_{i \neq t_x}\{F_{conv}^{ORTH}(x)_i\}). \quad (35)$$

For simplicity, adopt ReLU as activation, hence for a single layer:

$$F_l^{COCK}(x)_{t_x} - \max_{i \neq t_x}\{F_l^{COCK}(x)_i\} = \sigma(F_l^{ORTH}(x)_{t_x} - \max_{i \neq t_x}\{F_l^{ORTH}(x)_i\}). \quad (36)$$

Then for the $L$-layer neural network,

$$F^{COCK}(x)_{t_x} - \max_{i \neq t_x}\{F^{COCK}(x)_i\} = \sigma^L(F^{ORTH}(x)_{t_x} - \max_{i \neq t_x}\{F^{ORTH}(x)_i\}). \quad (37)$$

It holds

$$\Pr\{\sqrt{2}L_F^{COCK}||\epsilon||_2 \leq M_{F^{COCK},x}\} \geq \Pr\{\sqrt{2}L_F^{ORTH}\sigma^L||\epsilon||_2 \leq M_{F^{COCK},x}\}$$
$$= \Pr\{\sqrt{2}L_F^{ORTH}||\epsilon||_2 \leq M_{F^{ORTH},x}\}. \quad (38)$$

$\square$

## A.3 GENERALIZATION IN NORM

### A.3.1 GENERALIZATION FOR RISK AND ACCURACY

We first generalize Theorem 3.2.2 under the same setting in Theorem 3.2.2. Consider the case where $d_{\mathbb{X}}$ is defined by $l_p$ norm and $d_{\mathbb{Y}}$ is defined by $l_q$ norm.

**Theorem A.3.1.1.** *If $2^{1-\frac{1}{q}}L_F||\epsilon||_p \leq M_{F,x}$, then $M_{F,x+\epsilon} \geq 0$. That is, network $F$ is certified robust in $x$.*

*Proof.*

$$\Leftarrow 0 \leq F(x+\epsilon)_{t_x} - max_{i \neq t_x}\{F(x+\epsilon)_i\}$$

$$\Leftarrow F(x+\epsilon)_{t_x} - max_{i \neq t_x}\{F(x+\epsilon)_i\} \geq F(x)_{t_x} - max_{i \neq t_x}\{F(x)_i\} - 2^{1-\frac{1}{q}}L_F||\epsilon||_p$$

$$\text{LHS} = F(x+\epsilon)_{t_x} + F(x)_{t_x} - F(x)_{t_x} - max_{i \neq t_x}F(x)_i + max_{i \neq t_x}F(x)_i - max_{i \neq t_x}\{F(x+\epsilon)_i\}$$

$$= F(x)_{t_x} - max_{i \neq t_x}F(x)_i + F(x+\epsilon)_{t_x} - F(x)_{t_x} - [max_{i \neq t_x}\{F(x+\epsilon)_i\} - max_{i \neq t_x}F(x)_i]$$

$$\geq F(x)_{t_x} - max_{i \neq t_x}F(x)_i - |F(x+\epsilon)_{t_x} - F(x)_{t_x}| - max_{i \neq t_x}|F(x+\epsilon)_i - F(x)_i|$$

$$\geq F(x)_{t_x} - max_{i \neq t_x}F(x)_i - max_{a_1,a_2 \in \mathbb{R}}\{|a_1| + |a_2| \Big| (a_1^q + a_2^q)^{\frac{1}{q}} \leqslant L_F||\epsilon||_p\}$$

$$= F(x)_{t_x} - max_{i \neq t_x}F(x)_i - 2^{1-\frac{1}{q}}L_F||\epsilon||_p. \quad (39)$$

$\square$

Next, we generalize our definition of certified robust risk and accuracy.

**Theorem A.3.1.2.** *($\epsilon_p$-Margin Loss) Utilizing the definition of output margin in (7), the $\epsilon_p$-margin loss is any loss whose reduction leads to the enlargement of those output margins that less than $\epsilon$ in the sense of $l_p$ norm.*

The definition of empirical ($\epsilon_p$-)certified robust risk and expected ($\epsilon_p$-)certified robust risk can be simply generalized from traditional empirical risk and expected risk by replacing loss $l$ with ($\epsilon_p$-)margin loss.

Then, we will define the certified robust accuracy under the setting in Section 3.1 while $d_{\mathbb{X}}$ is defined by $l_p$ norm and $d_{\mathbb{Y}}$ is defined by $l_q$ norm.

**Theorem A.3.1.3. (Empirical Certified Robust Accuracy)**

$$ECR = \frac{1}{N}\sum_{i=1}^{N}(\mathbb{I}\{M_{F_\theta,x^{(i)}} \geq 2^{1-\frac{1}{q}}L_F||\epsilon||_p\}). \quad (40)$$

**Theorem A.3.1.4. (Expected Certified Robust Accuracy)**

$$ECR^* = \Pr\{M_{F_\theta,x} \geq 2^{1-\frac{1}{q}}L_F||\epsilon||_p\}. \quad (41)$$

### A.3.2 GENERALIZATION FOR COCK'S CERTIFIED ROBUSTNESS

The proof for Theorem 5.2.1 also holds for the case where $d_{\mathbb{X}}$ is defined by $l_p$ norm and $d_{\mathbb{Y}}$ is defined by $l_q$ norm, by simply generalizing the expected certified robust accuracy.

## A.4 OMITTED EXPERIMENTAL RESULTS

### A.4.1 ABLATION STUDY ON ONI STEPS

We investigate the effect of ONI steps $t$ using LipConvNet-15 on CIFAR-100 in Table 2. All the results are taken when the amount of distinct convolution kernel $k = 2$. In Table 2, the loss is the training loss and the certified robust accuracy is the performance during inference on the test set.

Table 2: Ablation On ONI Steps Using LipConvNet-15 On CIFAR-100

| $t$ | Loss | Training Accuracy(%) | Test Accuracy(%) | $\epsilon = \frac{36}{255}$ | $\epsilon = \frac{72}{255}$ | $\epsilon = \frac{108}{255}$ |
|---|---|---|---|---|---|---|
| 2 | 3.1825 | 24.16 | 24.21 | 13.91 | 7.24 | 3.33 |
| 3 | 2.6284 | 36.40 | 34.97 | 22.88 | 14.24 | 8.10 |
| 4 | 2.2366 | 45.87 | 41.21 | 29.27 | 18.79 | 11.70 |
| 5 | 2.0542 | 51.22 | 44.07 | 31.29 | 20.37 | 13.11 |
| 6 | 1.9926 | 53.08 | 44.63 | 32.03 | 21.14 | 13.31 |
| 7 | 1.9931 | 53.04 | 44.38 | 32.04 | 21.09 | 13.44 |
| 8 | **1.7111** | **57.92** | **46.24** | **35.19** | **25.47** | **17.58** |
| 9 | 2.1259 | 43.11 | 35.19 | 25.66 | 18.13 | 12.43 |

As ONI steps $t$ increases, the expressive power of convolution layers will be weakened, while the orthogonality of convolution layers will be strengthened which further improves the optimization condition. A carefully handcrafted ONI steps $t$ trades expressive power and optimization off to cover the training set better.

### A.4.2 TRAINING LOSS

We present the training loss in Figure 3. COCK covers the training set better than baseline orthogonal convolutions.

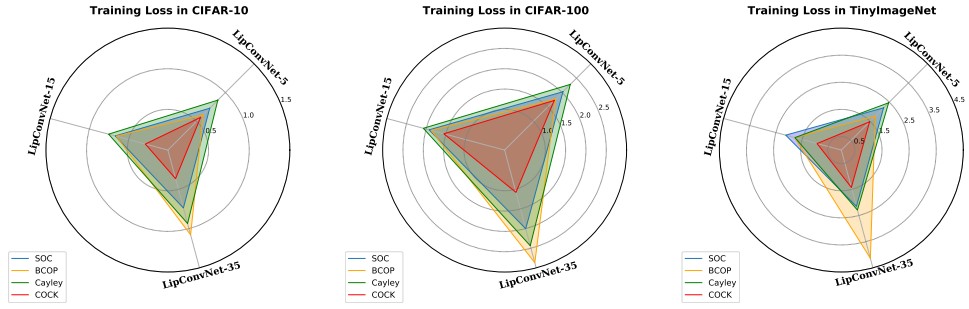

Figure 3: Training loss of different models in different datasets. The lower loss indicates that the model covers the training set better. The training loss of COCK (the red triangles) is lower than other orthogonalization methods consistently for datasets and models.

### A.4.3 COCK COVERS THE TRAINING SET

Results that suggest COCK covers the training set are shown in Table 3. The loss, accuracy and certified robustness in the table are all evaluated on the training set.

Table 3: COCK covers the training set in the case $k = 5$

| Model | Dataset | Loss | Accuracy(%) | $\epsilon = \frac{36}{255}$ | $\epsilon = \frac{72}{255}$ | $\epsilon = \frac{108}{255}$ |
|---|---|---|---|---|---|---|
| | **CIFAR-10** | 0.1434 | 97.05 | 92.86 | 84.92 | 72.82 |
| **LipConvNet-15** | **CIFAR-100** | 0.1337 | 98.41 | 96.32 | 92.06 | 83.89 |
| | **TinyImageNet** | 0.1515 | 99.41 | 97.77 | 92.96 | 81.27 |

### A.4.4 VANILLA CLASSIFICATION ACCURACY AND CERTIFIED ROBUST ACCURACY ON CIFAR-100

The results of vanilla classification accuracy and certified robust accuracy on CIFAR-100 are shown in Table 4.

Table 4: Vanilla Classification Accuracy and Certified Robust Accuracy on CIFAR-100

| Stage | Model | Method | Accuracy(%) | $\epsilon = \frac{36}{255}$ | $\epsilon = \frac{72}{255}$ | $\epsilon = \frac{108}{255}$ |
|---|---|---|---|---|---|---|
| **Training** | **LipConvNet-5** | SOC | 50.95 | 36.34 | 23.76 | 14.79 |
| | | BCOP | 60.50 | 44.13 | 28.95 | 17.46 |
| | | Cayley | 44.05 | 30.49 | 19.53 | 11.87 |
| | | COCK | **61.36** | **45.20** | **30.07** | **18.54** |
| | **LipConvNet-15** | SOC | 53.57 | 38.69 | 25.92 | 16.25 |
| | | BCOP | 56.57 | 40.48 | 26.56 | 16.19 |
| | | Cayley | 50.34 | 35.74 | 23.88 | 14.95 |
| | | COCK | **64.08** | **50.04** | **36.62** | **24.94** |
| | **LipConvNet-35** | SOC | 51.09 | 37.17 | 24.40 | 15.16 |
| | | BCOP | 30.81 | 19.04 | 11.09 | 6.18 |
| | | Cayley | 40.10 | 27.22 | 17.13 | 10.30 |
| | | COCK | **73.72** | **61.80** | **47.96** | **34.50** |
| **Test** | **LipConvNet-5** | SOC | **46.35** | **32.39** | 21.04 | 13.18 |
| | | BCOP | 46.00 | 31.71 | 20.33 | 12.26 |
| | | Cayley | 42.66 | 29.48 | 19.06 | 11.55 |
| | | COCK | 44.32 | 32.15 | **22.20** | **14.62** |
| | **LipConvNet-15** | SOC | **47.31** | 33.85 | 22.41 | 14.15 |
| | | BCOP | 43.90 | 30.16 | 20.12 | 12.37 |
| | | Cayley | 45.57 | 30.30 | 18.48 | 10.58 |
| | | COCK | 45.18 | **34.75** | **25.52** | **18.15** |
| | **LipConvNet-35** | SOC | **45.41** | **32.74** | 21.23 | 13.48 |
| | | BCOP | 30.37 | 19.25 | 11.02 | 6.22 |
| | | Cayley | 38.41 | 25.83 | 15.76 | 9.30 |
| | | COCK | 38.56 | 30.68 | **23.32** | **17.27** |

### A.4.5 VANILLA CLASSIFICATION ACCURACY AND CERTIFIED ROBUST ACCURACY ON TINYIMAGENET

The results of vanilla classification accuracy and certified robust accuracy on TinyImageNet are shown in Table 5.

### A.4.6 VANILLA CLASSIFICATION ACCURACY (BETTER) AND CERTIFIED ROBUST ACCURACY ON CIFAR-10

The results of vanilla classification accuracy and certified robust accuracy resulting from the trade-off for a better vanilla classification accuracy on CIFAR-10 are shown in Table 6.

Table 5: Vanilla Classification Accuracy and Certified Robust Accuracy on TinyImageNet

| Stage | Model | Method | Accuracy(%) | $\epsilon = \frac{36}{255}$ | $\epsilon = \frac{72}{255}$ | $\epsilon = \frac{108}{255}$ |
|---|---|---|---|---|---|---|
| | | SOC | 51.09 | 35.30 | 21.37 | 11.44 |
| | LipConvNet-5 | BCOP | 65.39 | 46.87 | 28.16 | 14.00 |
| | | Cayley | 45.46 | 29.92 | 17.31 | 9.07 |
| | | COCK | **66.87** | **52.67** | **37.90** | **25.15** |
| | | SOC | 52.69 | 36.92 | 22.91 | 12.70 |
| Training | LipConvNet-15 | BCOP | 64.19 | 44.19 | 26.32 | 12.48 |
| | | Cayley | 65.20 | 45.18 | 27.16 | 13.75 |
| | | COCK | **82.50** | **73.75** | **63.21** | **51.47** |
| | | SOC | 52.27 | 36.47 | 22.26 | 12.25 |
| | LipConvNet-35 | BCOP | 14.03 | 5.22 | 1.71 | 0.56 |
| | | Cayley | 49.78 | 32.16 | 18.16 | 8.86 |
| | | COCK | **69.55** | **53.39** | **36.12** | **21.05** |
| | | SOC | **36.10** | 23.90 | 14.76 | 8.77 |
| | LipConvNet-5 | BCOP | 33.89 | 21.96 | 12.89 | 7.46 |
| | | Cayley | 33.26 | 21.55 | 12.63 | 7.41 |
| | | COCK | 35.61 | **26.15** | **18.14** | **12.34** |
| | | SOC | **36.65** | 24.49 | 15.29 | 9.41 |
| Test | LipConvNet-15 | BCOP | 32.37 | 21.27 | 12.67 | 7.38 |
| | | Cayley | 33.36 | 22.06 | 12.76 | 7.33 |
| | | COCK | 35.90 | **30.19** | **24.31** | **19.61** |
| | | SOC | **34.91** | **23.13** | **14.26** | 8.24 |
| | LipConvNet-35 | BCOP | 12.35 | 4.83 | 2.02 | 0.75 |
| | | Cayley | 28.12 | 18.09 | 10.85 | 6.20 |
| | | COCK | 29.26 | 19.92 | 12.83 | **8.41** |

Table 6: Vanilla Classification Accuracy (Better) and Certified Robust Accuracy on CIFAR-10

| Stage | Model | Method | Accuracy(%) | $\epsilon = \frac{36}{255}$ | $\epsilon = \frac{72}{255}$ | $\epsilon = \frac{108}{255}$ |
|---|---|---|---|---|---|---|
| | LipConvNet-5 | COCK | 81.58 | 62.89 | 40.89 | 22.95 |
| Training | LipConvNet-15 | COCK | 83.60 | 71.54 | 57.39 | 43.36 |
| | LipConvNet-35 | COCK | 76.96 | 58.22 | 38.30 | 21.62 |
| | LipConvNet-5 | COCK | 76.05 | 58.86 | 40.93 | 25.51 |
| Test | LipConvNet-15 | COCK | 76.97 | 66.19 | 54.15 | 42.45 |
| | LipConvNet-35 | COCK | 72.17 | 54.91 | 37.07 | 22.62 |

### A.4.7 VANILLA CLASSIFICATION ACCURACY (BETTER) AND CERTIFIED ROBUST ACCURACY ON CIFAR-100

The results of vanilla classification accuracy and certified robust accuracy resulting from the trade-off for a better vanilla classification accuracy on CIFAR-100 are shown in Table 7.

Table 7: Vanilla Classification Accuracy (Better) and Certified Robust Accuracy on CIFAR-100

| Stage | Model | Method | Accuracy(%) | $\epsilon = \frac{36}{255}$ | $\epsilon = \frac{72}{255}$ | $\epsilon = \frac{108}{255}$ |
|---|---|---|---|---|---|---|
| | LipConvNet-5 | COCK | 54.87 | 38.14 | 24.39 | 14.47 |
| Training | LipConvNet-15 | COCK | 59.45 | 45.49 | 32.36 | 21.29 |
| | LipConvNet-35 | COCK | 53.17 | 38.00 | 25.31 | 15.59 |
| | LipConvNet-5 | COCK | 45.19 | 32.27 | 21.20 | 13.22 |
| Test | LipConvNet-15 | COCK | 46.24 | 35.19 | 25.47 | 17.58 |
| | LipConvNet-35 | COCK | 43.75 | 31.42 | 21.60 | 14.10 |

### A.4.8 VANILLA CLASSIFICATION ACCURACY (BETTER) AND CERTIFIED ROBUST ACCURACY ON TINYIMAGENET

The results of vanilla classification accuracy and certified robust accuracy resulting from the trade-off for a better vanilla classification accuracy on TinyImageNet are shown in Table 8.

Table 8: Vanilla Classification Accuracy (Better) and Certified Robust Accuracy on TinyImageNet

| Stage | Model | Method | Accuracy(%) | $\epsilon = \frac{36}{255}$ | $\epsilon = \frac{72}{255}$ | $\epsilon = \frac{108}{255}$ |
|---|---|---|---|---|---|---|
| | LipConvNet-5 | COCK | 62.49 | 49.12 | 35.94 | 24.32 |
| Training | LipConvNet-15 | COCK | 67.52 | 54.23 | 40.32 | 27.60 |
| | LipConvNet-35 | COCK | 54.69 | 37.40 | 22.02 | 11.46 |
| | LipConvNet-5 | COCK | 35.89 | 26.20 | 18.08 | 12.01 |
| Test | LipConvNet-15 | COCK | 36.18 | 27.24 | 19.89 | 14.28 |
| | LipConvNet-35 | COCK | 34.22 | 22.11 | 13.54 | 7.43 |

### A.4.9 EVALUATION ON THE TRADE-OFF BETWEEN POWER AND GENERALIZATION

To alter the expressive power of COCK, we adjust the combination $\hat{k} = (k_1, ..., k_L)$. For a certain combination of dataset and architecture, the larger $\hat{k}$ represents a powerful model. By handcrafting different $\hat{k}$, we evaluate the performance with respect to different expressive power. Given $k$ for a single layer under a certain combination of dataset and architecture, we also need to set $t$ corresponds to $k$. For simplicity, all layers under a certain combination of dataset and architecture follow the same rule to map $t$ with $k$. The mapping rule is shown in Table 9.

Table 9: Implementation Details

| Dataset | Model | $t$ | $k$ |
|---|---|---|---|
| | LipConvNet-5 | 7 | 1,2,...,9 |
| CIFAR-10 | LipConvNet-15 | 9 | 1,2,...,9 |
| | LipConvNet-35 | 7 | 1 |
| | | 5 | 2,3,...,9 |
| | LipConvNet-5 | 6 | 1,2,...,9 |
| CIFAR-100 | LipConvNet-15 | 8 | 1,2,...,9 |
| | LipConvNet-35 | 6 | 1 |
| | | 5 | 2,3,...,9 |
| | LipConvNet-5 | 9 | 1,2,...,9 |
| TinyImageNet | LipConvNet-15 | 9 | 1,2,...,9 |
| | LipConvNet-35 | 6 | 1,2,...,9 |

For the mapping from $k$ distinct kernels to $s^2$ convolution kernels, our strategy is rather simple. For each distinct kernel, we simply map it to an arbitrary kernel and some of its neighbours.

