# OpenReview forum: "Bridging The Gap Between Training and Testing for Certified Robustness"
_ICLR.cc/2025/Conference — Submitted to ICLR 2025_

### Official Review · Reviewer_R7ZA · 2024-11-03

**Soundness:** 2
**Presentation:** 2
**Contribution:** 2
**Rating:** 3
**Confidence:** 4

**Summary:**

The paper studies the gap between training and test certified accuracy and identifies a questionable trade-off between expressive power and generalization. Following this, the paper proposes a new orthogonal convolution named Controllable Orthogonal Convolution Kernel (COCK) which improves the certified robustness.

**Strengths:**

1. The paper studies an important problem of certifying robustness in neural networks.
2. The new convolution COCK learns parameters in the frequency domain which is novel and seems to improve the certified robustness.

**Weaknesses:**

The argument for the trade-off between training and test certified robustness raises some fundamental issues that are not discussed in the work (see Questions).

**Questions:**

### The fundamental questions I have are the following:
1. The trade-off between expressive power and generalization for certified robustness is questionable. It is primarily established based on the argument that "expressive power brings high generalization error to classification performance during inference (overfitting) and is certain to degrade the certified robustness on the test set" (lines 245-247). But, this doesn't hold for commonly used overparameterized neural networks, as they exhibit double descent phenomenon, that is, increasing the expressive power (near 0 training error) decreases the generalization error (test error) [P1].
2. Why the theoretical results established on the training set cannot extend to the test set from the same data distribution? The argument made by the authors for expecting a gap between the training and test certified robustness is "experimental neglect on the training set and the theoretical ignorance during inference on the test set" -- but, statistically when training and test sets are from the same distribution, the theory established on large training data would translate to the test set as well.

### Questions on COCK:
1. In Algorithm 1 (line 670), is the inverse Fourier transform in line 688 new compared to the existing orthogonal convolutions?
2. Line 295: it is not clear why $h=w=n$ and what is $n$?
3. Line 324: how does COCK provide data scaling? Please clarify it.
4. Line 330: in the toy example case, why is the comparison made against a diagonal matrix? Would COCK result in a diagonal matrix in any of the settings?
5. The toy example visualization is unclear. What exactly is visualized? Are the points in data space, or are they layer wise feature space? The feature space dimension is also not mentioned.

### Questions on experiments:
1. How COCK is different from SOC, BCOP, and Cayley, the orthogonal convolutions used in experiments? Please add citations for the other algorithms and briefly compare and contrast them.
2. The closest algorithm to COCK is layer-wise orthogonal training (LOT), as mentioned in line 116. However, a direct comparison is *not* made experimentally. Are the orthogonal convolutions used (SOC, BCOP, Cayley) do layer-wise orthogonalization?
3. what is max-min activation mentioned in line 383?
4. Line 393: $\rho$ is used for l2 norm perturbation. I think it is the same as $\epsilon$ used in sections before. Is it? If so, please follow the same notation for clarity.

Overall, I think the new convolution is novel and interesting that it improves the certified robustness compared to the other orthogonal convolutions. But, the premise of the trade-off analysis is not convincing.

[P1] Belkin et al. "Reconciling modern machine-learning practice and the classical bias–variance trade-off." PNAS 2019

---

> ### Author Response · Authors · 2024-11-23
>
> **The Fundamental Questions:**
>
> **Q1:** The trade-off between expressive power and generalization for certified robustness is questionable. It is primarily established based on the argument that "expressive power brings high generalization error to classification performance during inference (overfitting) and is certain to degrade the certified robustness on the test set" (lines 245-247). But, this doesn't hold for commonly used overparameterized neural networks, as they exhibit double descent phenomenon, that is, increasing the expressive power (near 0 training error) decreases the generalization error (test error) [P1].
>
> **A1:** In deep learning, overparameterized neural networks generalize well which is significantly different from traditional machine learning, induced by several implicit bias of optimization algorithms. In our work, we assume a well-conditioned optimization and mainly consider the expressive power. Hence, we draw an analogy to traditional machine learning.
>
> **Q2:** Why the theoretical results established on the training set cannot extend to the test set from the same data distribution? The argument made by the authors for expecting a gap between the training and test certified robustness is "experimental neglect on the training set and the theoretical ignorance during inference on the test set" -- but, statistically when training and test sets are from the same distribution, the theory established on large training data would translate to the test set as well.
>
> **A2:** In practice, the data sample size is limited, we do not consider the asymptotic behavior when the training data is sufficiently large.
>
> **Questions on COCK:**
>
> **Q1:** In Algorithm 1 (line 670), is the inverse Fourier transform in line 688 new compared to the existing orthogonal convolutions?
>
> **A1:** No, LOT utilizes the inverse Fourier transform in their algorithm.
>
> **Q2:** Line 295: it is not clear why $h=w=n$ and what is $n$?
>
> **A2:** In Appendix A.2.2., we merely assume input has the same height and weight and we denote it as $n$.
>
> **Q3:** Line 324: how does COCK provide data scaling? Please clarify it.
>
> **A3:** The singular values of COCK are in [0,1]. Any singular value $s\in(0,1)$ induces data scaling.
>
> **Q4:** Line 330: in the toy example case, why is the comparison made against a diagonal matrix? Would COCK result in a diagonal matrix in any of the settings?
>
> **A4:** We are considering the expressive power with the toy example. The key difference between COCK and standard orthogonal convolutions lies in the singular value distribution. The latter restricts the singular values to be strictly one while the former provides a broader range. For simplicity, we merely consider the simplest case (diagonal matrix) to exemplify the expressive power.
>
> **Q5:** The toy example visualization is unclear. What exactly is visualized? Are the points in data space, or are they layer wise feature space? The feature space dimension is also not mentioned.
>
> **A5:** As described in lines 326-331, the toy example is in the two-dimension space. The data points initially lies around the biggest circle (black). For orthgonal convolution networks, the four points still lie around the biggest circle after the transformation while COCK is able to transform points around the red ellipse which succeeds to classfy the XOR problem.
>
> **Questions on Experiments:**
>
> **Q1:** How COCK is different from SOC, BCOP, and Cayley, the orthogonal convolutions used in experiments? Please add citations for the other algorithms and briefly compare and contrast them.
>
> **A1:** The key difference lies in the fact that COCK provides a broader range of expressive power than the existing orthogonal convolutions, by adjusting its two hyperparameters, as described in Section 5. Citations for baselines are added in the new version.
>
> **Q2:** The closest algorithm to COCK is layer-wise orthogonal training (LOT), as mentioned in line 116. However, a direct comparison is _not_ made experimentally. Are the orthogonal convolutions used (SOC, BCOP, Cayley) do layer-wise orthogonalization?
>
> **A2:** Yes, those orthogonal convolutions are layer-wise. As an extension from LOT, COCK provides a broader range of expressive power than LOT. On the one hand, with fixed distinct convolution kernel $k$, COCK can recover the performance of LOT. On the other hand, COCK has both time complexity and space complexity better than LOT, as described in Section 2.
>
> **Q3:** What is max-min activation mentioned in line 383?
>
> **A3:** The max-min activation is adopted in SOC[1]. Please refer to SOC[1].
>
> **Q4:** Line 393:  is $\rho$ used for l2 norm perturbation. I think it is the same as $\epsilon$ used in sections before. Is it? If so, please follow the same notation for clarity.
>
> **A4:** Yes, I have modified the ambiguity in the new version.
>
> [1] Sahil Singla and Soheil Feizi. Skew orthogonal convolutions. In International Conferenceon Machine Learning, pp.9756–9766.PMLR,2021.

---

> > ### Comment · Reviewer_R7ZA · 2024-11-24
> >
> > Thank you for the response. I still find the fundamental questions not answered convincingly. Especially for the first question, the authors respond that the analogy is based on traditional machine learning methods. However, they use this analogy to reason for modern deep learning models, which are heavily overparameterized and are shown to deviate a lot from the traditional models. Thus, the fundamental concern still remains.

---

### Official Review · Reviewer_vFJT · 2024-11-06

**Soundness:** 3
**Presentation:** 3
**Contribution:** 3
**Rating:** 5
**Confidence:** 3

**Summary:**

This manuscript introduces the controllable orthogonal convolutional kernel (COCK), which uses a more flexible parameterization of convolution operation while controlling its Lipschitz constants. In this context, COCK demonstrates better flexibility while ensuring certified robustness.

**Strengths:**

1. The gap of certified robustness between the training and the test set is a key issue in provably robustness while underexplored. The proposed method is theoretically guaranteed.

2. The experiment is comprehensive.

3. The manuscript is generally well-written and easy to follow.

**Weaknesses:**

1. My major concern is the logic of the motivation and the proposed method. As Section 5 demonstrates, the proposed COCK is more flexible than vanilla orthogonal convolution, why doesn't it suffer more from overfitting? The last paragraph in Section 4.1 points out that there is a trade-off between expressive power (related to parameterization flexibility) and generalization.

2. Theorem 5.2.1 may need more explanation. I agree COCK demonstrates no-inferior performance in the experiment, but I am suspicious about the proof of this theorem. In the proof, especially Equation (35), the authors seem to assume that the margins of COCK and the baseline are the same, while it may not be the case. In addition, when using COCK for parameterization, there may exist no-inferior points compared with the baseline, but we cannot guarantee the optimization in training and lead the parameter to converge to such cases. In this regard, the conclusion of Theorem 5.2.1 looks too strong, the authors may need more assumptions or explanations to make this part more rigorous.

3. Although the experiment is comprehensive, the $\rho$ that we achieve good certified robustness is small, limiting the practical use of the method. Actually, in the empirical study of adversarial training, the $\epsilon$ value in the $l_2$ cases is usually set as $0.5$ or $2$, much larger than the values used in the experiment (e.g. Table 1) of this manuscript.

**Questions:**

The major questions are pointed out in the "weakness" part, the authors need to answer those questions to address my concerns.

Two minor issues:

1. In Definition 4.2.2, the difference between $\epsilon$ and $\epsilon_2$ seems unclear.

2. In Figure 1, the authors should distinguish the difference between the original input and the post-transformation features. In the current figure, it looks like there are $8$ points in total.

From my point of view, the current version of the manuscript is not ready for publication. I welcome the authors to address my concerns in the rebuttal.

---

> ### Author Response · Authors · 2024-11-23
>
> **Main Concerns:**
>
> **Q1:** My major concern is the logic of the motivation and the proposed method. As Section 5 demonstrates, the proposed COCK is more flexible than vanilla orthogonal convolution, why doesn't it suffer more from overfitting? The last paragraph in Section 4.1 points out that there is a trade-off between expressive power (related to parameterization flexibility) and generalization.
>
> **A1:** We describe the main logic and motivation as follows. By reviewing the current literature in certified robustness, we recognized there is a gap between training and testing for certified robustness (the experimental neglect on the training set and the theoretical ignorance during inference on the test set). We then identified a trade-off between expressive power and generalization for certified robustness by drawing an analogy from certified robustness to the overfitting and underfitting dynamics in traditional machine learning. To investigate this trade-off, we proposed COCK, providing a broader range of expressive power than existing orthogonal convolutions. On the one hand, COCK controls the singular values spectrum of the convolution kernel and alleviates the restricted orthogonal constraint to improve the expressive power. Notably, if setting a sufficiently large Newton iteration step $t$, COCK can recover the expressive power of standard orthogonal convolutions. On the other hand, by adjusting the amount of distinct convolution kernel $k$, COCK further controls the expressive power. The decreasing of $k$ degrades the expressive power, making COCK less powerful than standard orthogonal convolutions. In essence, the two hyperparameters $t$ and $k$ empower COCK with stronger flexibility in expressive power (COCK can either be more powerful or less powerful than standard orthogonal convolutions), making it an appropriate tool for investigating the trade-off for certified robustness.
>
> **Q2:** Theorem 5.2.1 may need more explanation. I agree COCK demonstrates no-inferior performance in the experiment, but I am suspicious about the proof of this theorem. In the proof, especially Equation (35), the authors seem to assume that the margins of COCK and the baseline are the same, while it may not be the case. In addition, when using COCK for parameterization, there may exist no-inferior points compared with the baseline, but we cannot guarantee the optimization in training and lead the parameter to converge to such cases. In this regard, the conclusion of Theorem 5.2.1 looks too strong, the authors may need more assumptions or explanations to make this part more rigorous.
>
> **A2:** Thanks for the helpful suggestions. The optimization is neglected in Theorem 5.2.1. In Theorem 5.2.1, we merely consider the problem of expressive power and prove from the parameter existence. To be specific, for Equation (35), it is rigorous to claim that "for a standard orthogonal convolution layer equipped with any parameters, there exists certain parameters for COCK such that Equation (35) holds." Indeed, we acknowledge this limitation in footnote 1 and in Section 7.1. We modify Theorem 5.2.1 in the new revision to avoid possible overclaims.
>
> **Q3:** Although the experiment is comprehensive, the $\rho$ that we achieve good certified robustness is small, limiting the practical use of the method. Actually, in the empirical study of adversarial training, the $\epsilon$ value in the $l_2$ cases is usually set as 0.5 or 2, much larger than the values used in the experiment (e.g. Table 1) of this manuscript.
>
> **A3:** In our experiments, we set the value of $\rho$ according to our baseline method SOC[1]. Actually, the perturbation is large enough for the reason that we normalize the perturbation according to the dimension of image (255). For larger perturbation such as $\rho=1,2$, the certified robust accuracy is low. Take LipConvNet-15 on CIFAR-100 as an example.
>
> * Training
>
>   |     | $\rho=1$ | $\rho=2$ |
>   | --- | --- | --- |
>   | BCOP | 0.62 | 0.0 |
>   | SOC | 0.756 | 2e-5 |
>   | Cayley | 0.5 | 0.0 |
>   | COCK | 3.112 | 0.042 |
>
> * Test
>
>   |     | $\rho=1$ | $\rho=2$ |
>   | --- | --- | --- |
>   | BCOP | 0.98 | 0.0 |
>   | SOC | 1.25 | 0.0 |
>   | Cayley | 0.82 | 0.0 |
>   | COCK | 3.04 | 0.09 |
>
>
> **Minor Issues:**
>
> **Q1:** In Definition 4.2.2, the difference between $\epsilon$ and $\epsilon_2$ seems unclear.
>
> **A1:** In Definition 4.2.2, we define $\epsilon_2$-margin for margin within $l_2$ norm, where $\epsilon_2$-margin is just a notation and $\epsilon$ in the Definition is some specific perturbation magnitude.
>
> **Q2:** In Figure 1, the authors should distinguish the difference between the original input and the post-transformation features. In the current figure, it looks like there are 8 points in total.
>
> **A2:** We clarify the ambiguity in the paragraph (line335-line348).
>
> [1] Sahil Singla and Soheil Feizi. Skew orthogonal convolutions. In International Conferenceon Machine Learning, pp.9756–9766.PMLR,2021.

---

> > ### Comment · Reviewer_vFJT · 2024-11-25
> > **Update**
> >
> > I thank the authors for their responses. After checking the rebuttal and the reviews from other reviewers, I decided to keep my current score unchanged. The robust accuracy of COCK is not competitive while the motivation or the logic (as mentioned in my first point and some other reviewers) are not convincing, either.

---

### Official Review · Reviewer_ck3u · 2024-11-07

**Soundness:** 1
**Presentation:** 1
**Contribution:** 1
**Rating:** 1
**Confidence:** 4

**Summary:**

This paper designs a new certified robustness method named COCK that is claimed to enjoy a better robustness-utility trade-off (but I don't understand what is the intuition behind the design of COCK). Experiments are conducted on CIFAR-10, CIFAR-100, and Tiny-ImageNet to evaluate the proposed COCK method (which shows that COCK is significantly worse than most recent certified robustness methods).

**Strengths:**

None.

**Weaknesses:**

1. The certified robustness performance of the proposed COCK method is significantly weaker than recent certified robustness methods such as [r1] and [r2]. Specifically, for example, according to Table 1, when the $l\_2$-perturbation radius is $\frac{108}{255}$, the best certified robust accuracy COCK achieved on CIFAR-10 dataset is $60.79\\%$. However, with a higher perturbation radius $\rho = 0.5 > \frac{108}{255}$, on the CIFAR-10 dataset, [r1] and [r2] achieved certified robust accuracies of $65.5\\%$ and $70.7\\%$ respectively, which both are significantly higher than the result of COCK. Similar things also happen for the other dataset (CIFAR-100) and other perturbation radii. Furthermore, [r1] and [r2] can work on the ImageNet dataset, while COCK is only analyzed on the Tiny-ImageNet dataset. So if the proposed COCK method is so weak, why the community needs to waste time on it?

2. Algorithm 1 (i.e., COCK) seems to be very time-consuming, which may significantly limit the practicality of the proposed COCK method. There are two main calculation challenges: (1) Line#681 needs to conduct $3t$ times of matrix multiplication (i.e., calculate the term $B^3\_{\tau-1} S$ for $t$ times), and (2) Line#688 needs to perform inverse-FFT for $i*j$ times. So I suggest the authors analyze the theoretical and empirical time complexity of COCK. Also please note that [r1] and [r2] can even work without additional model training.

3. The authors claim that one of their contributions is that they discover "a trade-off between expressive power and generalization for certified robustness" (For example, see Line#078-079). Unfortunately, they are not the first to find the trade-off and such an observation is now very trivial. Specifically, the trade-off between model robustness and utility has been widely accepted by the ML S&P community and has also been well studied in many existing works such as [r3], [r4], [r5], and [r6].

4. I still do not understand what is the motivation for designing COCK? How can it make COCK enjoys certified robustness better than existing certified robustness approaches (although it actually cannot)? Please comment.

5. Experiments could be significantly improved. Firstly, I suggest including experiments on the (vanilla) ImageNet dataset. Secondly, I think the authors should cite more recent certified robustness baselines and compare them with your proposed COCK. It is worth noting that the most recent paper that the authors cited in this work is from 2022, which is still a too-old paper (in the field of certified robustness).

**References:**

[r1] Carlini N. et al. "(Certified!!) Adversarial Robustness for Free!" arXiv 2206.10550 / ICLR 2023.

[r2] Chen H. et al. "Diffusion Models are Certifiably Robust Classifiers." arXiv 2402.02316 / NeurIPS 2024.

[r3] Raghunathan A. et al. "Understanding and Mitigating the Tradeoff between Robustness and Accuracy." ICML 2020.

[r4] Donhauser K et al. "Interpolation can hurt robust generalization even when there is no noise." NeurIPS 2021.

[r5] Rice L. et al. "Overfitting in adversarially robust deep learning." ICML 2020.

[r6] Fu S. et al. "Theoretical Analysis of Robust Overfitting for Wide DNNs: An NTK Approach." ICLR 2024.

**Questions:**

See **Weaknesses** for details.

---

> ### Author Response · Authors · 2024-11-23
>
> **Main Questions:**
>
> **Q1:** The certified robustness performance of the proposed COCK method is significantly weaker than recent certified robustness methods such as [r1] and [r2]. Specifically, for example, according to Table 1, when the $l_2$-perturbation radius is $\frac{108}{255}$ , the best certified robust accuracy COCK achieved on CIFAR-10 dataset is $60.79\%$. However, with a higher perturbation radius $\rho=0.5>\frac{108}{255}$ , on the CIFAR-10 dataset, [r1] and [r2] achieved certified robust accuracies of $65.6\%$ and $70.7\%$ respectively, which both are significantly higher than the result of COCK. Similar things also happen for the other dataset (CIFAR-100) and other perturbation radii. Furthermore, [r1] and [r2] can work on the ImageNet dataset, while COCK is only analyzed on the Tiny-ImageNet dataset. So if the proposed COCK method is so weak, why the community needs to waste time on it?
>
> **A1:** The experiment setting in [r1] and [r2] is totally different from ours in terms of network architecture and method. We use LipConvNet equipped with orthogonal convolutions, while both the two papers consider the diffusion methods/models. The comparison is not fair.
>
> **Q2:** Algorithm 1 (i.e., COCK) seems to be very time-consuming, which may significantly limit the practicality of the proposed COCK method. There are two main calculation challenges: (1) Line#681 needs to conduct $3t$ times of matrix multiplication (i.e., calculate the term $B_{\tau-1}^3S$ for $t$ times), and (2) Line#688 needs to perform inverse-FFT for $i*j$ times. So I suggest the authors analyze the theoretical and empirical time complexity of COCK. Also please note that [r1] and [r2] can even work without additional model training.
>
> **A2:** Firstly, the motivation and role of COCK is to provide a broader range of expressive power, serving as a tool to investigate the trade-off between expressive power and generalization for certified robustness. Secondly, in terms of the time complexity, it is not fair to directly compare COCK with those in [r1] and [r2] because the setting is totally different. We empirically compare the training time of COCK with our baseline methods in the case of LipConvNet-25 training on CIFAR-100. Compared with our baselines, COCK is not time-consuming.
>
> |     | Training time per epoch / (s) |
> | --- | --- |
> | SOC | 642 |
> | BCOP | 349 |
> | Cayley | 204 |
> | COCK | 234 |
>
> **Q3:** The authors claim that one of their contributions is that they discover "a trade-off between expressive power and generalization for certified robustness" (For example, see Line#078-079). Unfortunately, they are not the first to find the trade-off and such an observation is now very trivial. Specifically, the trade-off between model robustness and utility has been widely accepted by the ML S&P community and has also been well studied in many existing works such as [r3], [r4], [r5], and [r6].
>
> **A3:** The literature [r3] mainly uncovers the discrepancy between vanilla classification accuracy and adversarial robustness, which belongs to our summary of related works. We have appended this literature to our new version. The literature [r4][r5][r6] is different from ours. Mainly, they discuss robust overfitting in the setting of adversarial training. They characterize and understand the robust overfitting behavior from the robust risk. In our setting, however, we consider the standard training case and characterize the adversarial robustness by certified robustness. More importantly, the key difference is that our claim "a trade-off between expressive power and generalization for certified robustness" actually considers the impact of varying expressive power, while in those literature[r3][r4][r5], they consider a certain model training for a long time and studied its overfitting behavior.
>
> **Q4:** I still do not understand what is the motivation for designing COCK? How can it make COCK enjoys certified robustness better than existing certified robustness approaches (although it actually cannot)? Please comment.
>
> **A4:** We describe the main logic and motivation as follows. By reviewing the current literature in certified robustness, we recognized there is a gap between training and testing for certified robustness (the experimental neglect on the training set and the theoretical ignorance during inference on the test set). We then identified a trade-off between expressive power and generalization for certified robustness by drawing an analogy from certified robustness to the overfitting and underfitting dynamics in traditional machine learning. To investigate this trade-off, we proposed COCK which provides a broader range of expressive power than existing orthogonal convolutions.
>
> COCK guarantees a smaller Lipschitz norm and greater expressive power compared with standard orthogonal convolutions, hence enjoys better certified robustness (assuming a well-conditioned optimization). Please refer to Theorem 5.2.1.

---

> ### Comment · Reviewer_ck3u · 2024-11-23
> **I vote for strong rejection**
>
> Since the authors' rebuttal address none of my concern, I strongly vote for rejecting this paper. Detailed comments are as follows:
>
> **About Q1:** So why don't you apply your COCK method to more practical real-world models such as ResNets or ViTs rather than your toy LipConvNet? Please note that ResNets and ViTs are widely used models in the field of certified robustness. Is it becuase your COCK is so inefficient that it can not be applied to these real-world models?
>
> **About Q2:** So what is the theoretical time-complexity of COCK exactly? You are not answering my question. Besides, you should also note that your presented table clearly show that your method is not practical. It indicates that if you want to train a LipConvNet for 100 epochs, you will need 6.5 hours. I have never used such a long time on ResNet-18 with the standard adversarial training.
>
> **About Q3:** It seems like even you also agree that your paper is re-discovering common knowledge in [r3].
>
> **About Q4:**
> > *... We then identified a trade-off between expressive power and generalization for certified robustness by drawing an analogy from certified robustness to the overfitting and underfitting dynamics in traditional machine learning.*...
>
> Again, you did not identify anything new. You are just re-discovering common knowledge from existing papers such as [r3].
>
> **About Q5:** You ignored and refused to answer my question.

---

### Official Review · Reviewer_SoVx · 2024-11-08

**Soundness:** 3
**Presentation:** 2
**Contribution:** 2
**Rating:** 5
**Confidence:** 2

**Summary:**

This paper aims to understand the differences in certified robustness observed during training and testing phases, especially focusing on how the expressive power of models impacts certified robustness on the test set.

The authors identify a trade-off between expressive power and generalization for certified robustness, drawing an analogy to the overfitting and underfitting dynamics in traditional machine learning. To explore this, they introduce a new model component, the Controllable Orthogonal Convolution Kernel (COCK), which allows for greater flexibility and a broader range of expressive power than existing orthogonal convolution methods.

**Strengths:**

1. This paper provides a fresh perspective by drawing a parallel between the traditional machine learning concepts of underfitting and overfitting and the balance between expressive power and generalization in certified robustness. This analogy sheds light on an often-overlooked aspect in the field, emphasizing that certified robustness exhibits similar trade-offs as seen in conventional learning. This insight could potentially open new avenues for the research community to deepen their understanding of certified robustness.

2. The authors introduce the Controllable Orthogonal Convolution Kernel (COCK), a module specifically designed to adjust the expressive power of neural networks. By enabling more nuanced control over the model’s expressive capacity, COCK serves to bridge the gap between training and test robustness.

**Weaknesses:**

1. The central finding of the paper—that there exists a trade-off between expressive power and generalization for certified robustness— is not surprising and may be considered common knowledge in this field. While the paper empirically validates this trade-off, it lacks both theoretical understanding and empirical analysis of this trade-off.

2. Although the paper introduces new definitions and theorems, these contributions primarily reinterpret existing ideas rather than advancing theoretical understanding. For instance, Proposition 5.1.1 on COCK appears to be more of an application or extension of existing techniques rather than a novel theoretical innovation. Thus, the paper’s contributions seem limited.

**Questions:**

See the Weaknesses.

Overall, I think this paper raises an interesting phenomenon, but the theoretical research or experimental analysis of this phenomenon is not deep enough and needs further improvement.

---

> ### Author Response · Authors · 2024-11-23
>
> **Main Concerns:**
>
> **Q1**: The central finding of the paper—that there exists a trade-off between expressive power and generalization for certified robustness— is not surprising and may be considered common knowledge in this field. While the paper empirically validates this trade-off, it lacks both theoretical understanding and empirical analysis of this trade-off.
>
> **A1**: By investigating certified robustness literature and recognizing the experimental neglect on the training set and the theoretical ignorance during inference on the test set for certified robustness, this paper throws out a significant problem—there exists a gap between training and testing for certified robustness. Typically, how to improve the certified robustness on the test set is unclear. Further, this paper provides a fresh perspective by drawing a parallel between the traditional machine learning concepts of underfitting and overfitting and the balance between expressive power and generalization in certified robustness. Overall, the intuitively trivial but significant gap and parallel are worth emphasizing for the certified robustness community to understand the certified robustness on the test set better.
>
> Empirically, we not only empirically validate this trade-off, but also raise an interesting power-driven shift phenonmenon using a new-designed orthogonal convolution COCK. This phenonmenon further uncovers how this trade-off for certified robustness behaves.
>
> **Q2**: Although the paper introduces new definitions and theorems, these contributions primarily reinterpret existing ideas rather than advancing theoretical understanding. For instance, Proposition 5.1.1 on COCK appears to be more of an application or extension of existing techniques rather than a novel theoretical innovation. Thus, the paper’s contributions seem limited.
>
> **A2**: Most of the new definitions and theorems in the paper aim to either validate the trade-off between expressive power and generalization in certified robustness or establish the parallel between the traditional machine learning concepts of underfitting and overfitting and the balance between expressive power and generalization in certified robustness. Though the definitions or theorems for the parallel are trivial extensions from basic machine learning, they are fundamental and significant for further study of certified robustness on the test set.
>
> Proposition 5.1.1 is indeed an extension of existing techniques and we do not view this proposition as our contribution.

---

> > ### Comment · Reviewer_SoVx · 2024-11-25
> > **Acknowledgement**
> >
> > I thank the responses from the authors. I will remain my scores, too.

---

### Official Review · Reviewer_ukL7 · 2024-11-11

**Soundness:** 2
**Presentation:** 1
**Contribution:** 1
**Rating:** 3
**Confidence:** 3

**Summary:**

This paper presents an analysis on certified robustness on Convolutional Netural Networks (CNNs). The authors first show an equivalence between the convergence of training loss and improvement of certified robustness. After that, a new orthogonal convolution-Controllable Orthogonal Convolution Kernel (COCK) is proposed to mitigate the trade-off between expressive power and generalization. Through experiments on CIFAR10 dataset, the authors show that COCK achieves improvement on vallina accuracy and certified accuracy compared to existing baselines.

**Strengths:**

The proposed method performs better than the baseline on most cases.

**Weaknesses:**

I believe the current paper is not in a publishable quality for the following reasonings:
1. Preliminary is not clear. In Section 3, the definition of cerfified robustness is missing. It's also not well-explained how it's connected to orthogonal convolution and Lipschitz-contrainted models.
2. Definition is not clear, notation is not consistent. For example, in the beginning of Section 3, $x^{(i)}$ denotes the $i$-th example in the training set, while in Equation (10), $x^{(ij)}$ denotes the $j$-th example that belongs to class $i$. The definitions of Convolition (3.3.1), margin loss (4.2.1) and certified robust accuracy (4.2.3 and 4.2.4) are also not clear.
3. Theorem 4.1.1 is not clear. I don't think it trivial that "minimizing training loss is equivalent to maximizing the output margin $m_k^{(ij)}$". It's also not clear how equivalent it is between the derease of training loss and the improvement of certified robustness. One should give a mathematical description to support the argument.
4. The proposed algorithm COCK is not well explained. How it it's related to Proposition 5.1.1 (and Fourier Transform)？What does COCK actually do? There is not adequate explanation on section 5.
5. How COCK is related to r-RELU? Also, Figure 1 is not well captioned.
6. The experiments are limited to CIFAR10 dataset and LipConvNet. Also, the citation of baselines (SOC, BCOP and Cayley) are missing.

**Questions:**

Please refer to the "Weakness" above.

---

> ### Author Response · Authors · 2024-11-23
>
> **Main Questions:**
>
> **Q1:** Preliminary is not clear. In Section 3, the definition of cerfified robustness is missing. It's also not well-explained how it's connected to orthogonal convolution and Lipschitz-contrainted models.
>
> **A1:** In Section 3, the definition of certified robustness is included in Theorem 3.2.2. As described in lines 186-192, based on the definition of certified robustness, one can regularize the spectrum (singular values) of the linear weight to achieve a Lipschitz-constrained neural network. That is because the Lipschitz norm of the linear layer is exactly the maximum singular value of the linear weight. A well-known technique to regularize the spectrum is to impose layer-wise weight regularization. Typically, by leveraging layer-wise orthogonal constraints, the Lipschitz norm of the linear layer is 1. In this way, we connect the definition of certified robustness to orthogonal convolution and Lipschitz-constrained models.
>
> **Q2:** Definition is not clear, notation is not consistent. For example, in the beginning of Section 3, $x^{(i)}$ denotes the $i$-th example in the training set, while in Equation (10), $x^{(ij)}$ denotes the $j$-th example that belongs to class $i$. The definitions of Convolition (3.3.1), margin loss (4.2.1) and certified robust accuracy (4.2.3 and 4.2.4) are also not clear.
>
> **A2:** As described in **Q2**, $x^{(i)}$ denotes the $i$-th example in the training set, while in Equation (10), $x^{(ij)}$ denotes the $j$-th example that belongs to class $i$. These two notations are clear. Definition 3.3.1 defines the orthogonal convolution kernel and orthogonal convolution neural networks. Definition 4.2.3 and Definition 4.2.4 define the certified robust accuracy on the training set and test set respectively.
>
> **Q3:** Theorem 4.1.1 is not clear. I don't think it trivial that "minimizing training loss is equivalent to maximizing the output margin $m_k^{(ij)}$". It's also not clear how equivalent it is between the derease of training loss and the improvement of certified robustness. One should give a mathematical description to support the argument.
>
> **A3:** By Equation (11), the training loss is monotone decreasing with respect to the output margin $m_k^{(ij)}$. Trivially, minimizing training loss is equivalent to maximizing the output margin $m_k^{(ij)}$.
>
> **Q4:** The proposed algorithm COCK is not well explained. How it it's related to Proposition 5.1.1 (and Fourier Transform)？What does COCK actually do? There is not adequate explanation on section 5.
>
> **A4:** In essence, COCK extends the Orthogonal Newton Iteration (ONI) algorithm to convolutions. ONI reparameterizes an orthogonal matrix $W$ by $W=(VV^T)^{-\frac{1}{2}}V$ and leverages Newton Iteration to iteratively approximate $(VV^{-\frac{1}{2}})$. In the setting of convolution, by proposition 5.1.1, we intend to orthogonalize the convolution layer via orthogonalizing each $P^{(u,v)}$. The overall algorithm is shown in Appendix A.1.
>
> **Trick in COCK:** To provide a broader range of expressive power, we share the parameters of some convolution kernels by handcrafting the amount of distinct convolution kernel $k$. Details are shown in lines 310-315.
>
> **Q5:** How COCK is related to r-RELU? Also, Figure 1 is not well captioned.
>
> **A5:** In Figure 1, we define r-RELU to exemplify the expressive power of COCK. In this example, standard orthogonal methods fail to solve the XOR binary classification problem but COCK succeed to solve the problem, as shown in Figure 1. We modify the caption of Figure 1 for clarification in the new revision.
>
> **Q6:** The experiments are limited to CIFAR10 dataset and LipConvNet. Also, the citation of baselines (SOC, BCOP and Cayley) are missing.
>
> **A6:** Our experiments are not limited to the CIFAR10 dataset, but also include the CIFAR100 and the TinyImageNet dataset as described in lines 402-404. Results are shown in Appendix A.4.4 and Appendix A.4.5. We add the citation of baselines in the new revision. Thanks for the helpful suggestion.

---

> > ### Comment · Reviewer_ukL7 · 2024-11-24
> > **Acknowledgement**
> >
> > I thank the responses from the authors. I will remain my scores.

---

### Meta-Review · Area_Chair_Fn5v · 2024-12-17

**Metareview:**

Summary: This paper studies the problem of trade-off between expressive power and generalization for certified adversarial robustness. The paper makes an analogy with overfiting and underfiting in conventional machine learning and proposes a new technique named Controllable Orthogonal Convolution Kernel (COCK). Experiments on the CIFAR10/100 and TinyImageNet are shown to test the proposed method.

Strength:
1. The problem of study is important and might be of broad interest to the adversarial robustness community.
2. The proposed method COCK performs well on the CIFAR10/100 and TinyImageNet datasets.

Limitation:
1. The paper is not well-written. Many definitions, concepts and notations are not well presented. The logic of the motivation and the proposed method is unclear to the reviewers.
2. Reviewers are concerned about the novelty of the paper. It seems that the techniques are not new and the contributions are limited. The trade-off between robustness and utility has been well studied in the previous literature.
3. Experiments are not comprehensive. Reviewers encourage the authors to do an experiment on the (vanilla) ImageNet dataset and compare with more recent baselines on certified robustness.
4. Reviewers have fundamental concerns about the analogy in the paper with conventional machine learning. Reviewers believe over-parametric deep neural networks should have a different analysis with conventional machine learning.

Given all reviewers vote for a rejection of the paper consistently, AC would follow the reviewers' opinions. AC encourages the authors to take the limitations above into account in the next version.

**Additional Comments On Reviewer Discussion:**

The reviewers have written a rebuttal in response to reviewers' comments in the first round of review. Unfortunately, reviewers comment that the rebuttal is not very convincing and they are not willing to change their scores. Specifically, reviewers have the following common comments:
1. The paper is not well-written (concepts, definitions, notations, etc.) and the motivation is unclear.
2. Reviewers have fundamental concerns on the novelty and contribution made in this paper.
3. Reviewers believe the experiments are not comprehensive.

After discussions, reviewers all believe the paper is below the ICLR bar and vote for (strong) rejection consistently.

---

### Decision · Program_Chairs · 2025-01-22

Reject